

# Geomorphic signature of relief rejuvenation in Sierra Morena (Betic forebulge, Spain): evidence of segmented uplift in a strongly strain-partitioned, tectonic scenario

Inmaculada Expósito[1]♦, Alejandro Jiménez-Bonilla[1]♦, Michele Delchiaro[2]♦, José L. Yanes[1], Juan C. Balanyá[1], Francisco Moral-Martos[1], Marta Della Seta[2]

[1]Departamento de Sistemas Físicos, Químicos y Naturales, Universidad Pablo de Olavide, Seville, 41013, Spain
[2]Department of Earth Sciences, Sapienza University of Rome, Rome, 00185, Italy

♦These authors contributed equally to this work

*Correspondence to*: Inmaculada Expósito (iexpram@upo.es)

**Abstract.** The foreland relief of Alpine orogenic belts is often rejuvenated due to the intraplate propagation of orogenic deformation. Thus, in these long-lived areas, the localisation of relief rejuvenation may be largely controlled by the reactivation of previous mechanical discontinuities. In this regard, we explored the relationship between the relief rejuvenation pattern and the distribution, geometry and kinematics of faults in a wide segment of the Betics foreland (Sierra Morena, Southern Spain). Specifically, we focused on the forebulge, a WSW-ENE flexural relief that formed, paired to the Betics foreland basin, in response to orogenic load. For this purpose, we applied both qualitative and quantitative geomorphological tools, including geomorphic indexes and knickpoint pattern modelling in χ space. We found that the pattern of relief rejuvenation responds to the tectonic activity of two groups of faults that often show evidence of reactivation: overall WSW-ENE faults contributing to both regional NNW-SSE relief segmentation and vertical extrusion of the forebulge, and NW-SE reverse faults associated with an outstanding WSW-ENE topographic segmentation in the west of the study area. In addition, our knickpoint modelling suggests that the faults related to the southernmost Sierra Morena mountain front have been particularly active in recent times, although their activity span and the relative uplift that they accommodate differ along the Sierra Morena/foreland basin limit. The knickpoint pattern also suggests a significant reorganisation of the analysed drainage basins. The strain partitioning accommodated by the structures involved in relief rejuvenation suggests the intraplate propagation of the transpressional deformation reported from the Betics external fold and thrust belt.

## 1 Introduction

Relief rejuvenation of geologically ancient areas often occurs in response to intraplate deformation. In these long-lived areas, reactivation of pre-existing structures is a common mechanism for strain accommodation, which is often favoured over the generation of new structures (e.g., Butler et al., 1997). Thus, previous discontinuities can be expected to localise the relief rejuvenation associated with recent or active tectonics, which can therefore be analysed to delve into the kinematics of recent deformation.





The relief of the Variscan basement outcropping in the western and central Iberian Peninsula is characterised by roughly W-E or WSW-ENE topographic highs (1,000 to 2,500 m a.s.l.) separated by Cenozoic sedimentary basins (Fig. 1a). On the basis of both numerical and analogue models, these topographic domains have been mainly interpreted as the expression of lithospheric

folding (Cloethingh et al., 2002; Fernández Lozano et al., 2012) due to the superposition of successive intraplate deformation phases from the Late Cretaceous-Early Cenozoic onwards. They have been mainly associated with plate-boundary activity both to the north (Pyrenean-Cantabrian orogen; e.g., Roest and Srivastava, 1991; Teixell., 2018) and to the south (Gibraltar Arc; e.g., Spakman, W. and Wortel, R., 2004; Balanyá et al., 2012) of Iberia (Fig.1a). In the upper crust, such intraplate deformation is often accommodated by reactivation of pre-existing faults, which largely control the spatial distribution and

orientation of the topographic features (De Vicente et al., 2007; Martín Gonzalez, 2009; Brum da Silveira et al., 2009; Vázquez-Vilchez et al., 2015).

The southernmost topographic high (900–1,300 m a.s.l.) within the Variscan Iberia is Sierra Morena (Fig. 1a, b). It is a WSE-ENE relief, approximately 600 km-long and 80 km-wide, WSW-ENE relief, the southern limit of which coincides with a regional linear escarpment separating the foreland (i.e., the Variscan Iberian Massif) from the Cenozoic Guadalquivir foreland

basin of the Betic Chain (northern branch of the Gibraltar Arc). Sierra Morena as a whole is interpreted as a forebulge, a positive relief that develops, paired with foreland basins subsidence, due to lithospheric flexure in front of major fold-and-thrust belts (e.g., Flemings and Jordan, 1990; DeCelles, 2012, and references therein). In this tectonic context, the Sierra Morena forebulge uplift is linked to the lithospheric flexure of the Iberian Massif under the Betics orogenic wedge load, enhanced by the buckling of the Iberian lithosphere (Cloething et al, 2002; García-Castellanos et al., 2002). Concerning this

folding, a NW-SE-oriented compression, connected to the convergence between Africa and Eurasia, is invoked for the neotectonic intraplate deformation of Iberia (Herraiz et al., 2000; Cloething et al, 2002).

This interpretation of the overall Sierra Morena relief as a flexural forebulge folded by NW-SE-oriented buckling is mainly based on the numerical modelling of the lithosphere rheological behaviour (Cloething et al, 2002; García-Castellanos et al., 2002). Nevertheless, little work has been done to address the geomorphological imprint of this tectonic model in Sierra Morena.

In this regard, previous studies propose that the tectonic setting for the current deformation of the western Betics fold-and-thrust belt (FTB) includes the westward migration of the Gibraltar Arc hinterland (Balanyá et al., 2012; Gutscher et al., 2002). This would imply that the tectonic mechanism responsible for the Miocene accretion of the Betics (e.g., Faccenna et al., 2004; Balanyá et al., 2007) is still active. Additionally, analytical and analogue models (Díaz-Azpiroz et al., 2014; Barcos et al., 2015, 2016) provide a N99°E–N109°E trending horizontal velocity vector to explain the Neogene to recent dextral

transpressional kinematics observed in segments of the western Betics FTB (see also García et al., 2016; Jiménez-Bonilla et al., 2015). This velocity vector orientation fits well with such westward motion of the hinterland relative to the FTB.

Irrespective of the current tectonic setting, Neogene, or even younger activity has been reported in specific areas of Sierra Morena. Thus, in the Guadiamar drainage basin (westernmost Sierra Morena, GB in Fig. 1c), river migration due to Holocene regional tilting has been described (Salvany, 2004). Additionally, seismicity consistent with reactivation of pre-existing

Earth **Surface**
**Dynamics**
Discussions

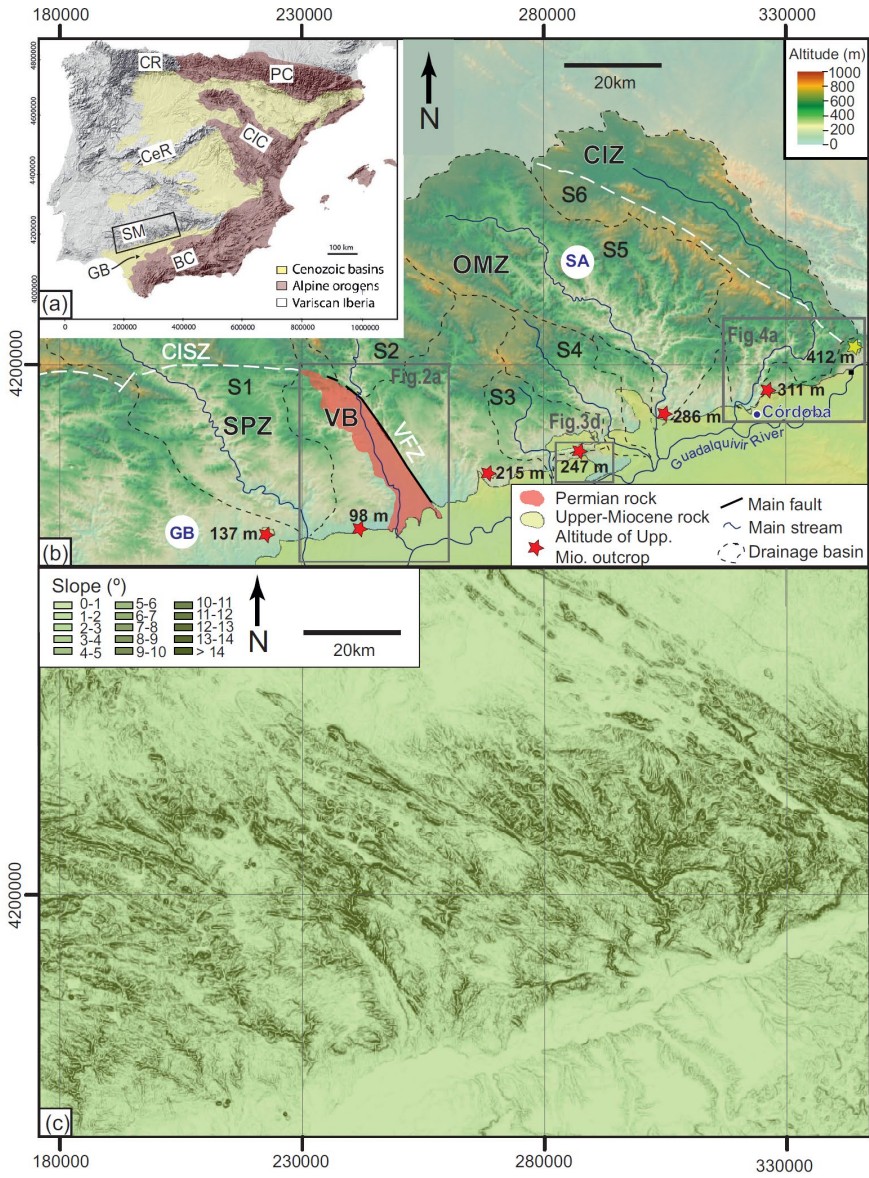

**Fig.1. Location and topography of the study area. (a) Topographic domains of the Iberian Peninsula related to the main geological units: BC, Betic Chain; GB, Guadalquivir basin; SM, Sierra Morena; CeR, Central Range; CR, Cantabrian Range; CIC, Central Iberian Chain, PC, Pyrenean Chain. (b) Altitude map of the study area (see location on Fig.1a); S1 to S6: drainage basins analysed in this study (see text for explanation); OMZ, Ossa-Morena Zone; SPZ, South Portuguese Zone; VB, Permian Viar Basin; VFS, Viar fault system; SISZ, South Iberian Shear Zone; GB. Guadiamar basin; SA, Sierra Albarrana. (c) Slope maps of the same area; note the preservation of low-slope relief surfaces on the interfluves of Sierra Morena. Maps in (b) and (c) are extracted from the m DEM 12 m resolution TanDEM-X DEM (Wessel, 2016). The reference system is WGS84 UTM 30N.**



fractures has been pointed out in the Sierra Albarrana area (SA in Fig. 1c; Herraiz et al., 1996), and the Upper Miocene infill

of the Guadalquivir foreland basin appears locally intersected by faults (Expósito et al., 2016; Matas et al., 2010 a).

In this study, we assume that this recent tectonic activity, linked to intraplate tectonics, must have rejuvenated the Sierra Morena relief. Therefore, its geomorphological features must be an expression, among other factors, of its current tectonic setting. Furthermore, we expect that the Sierra Morena relief rejuvenation pattern is significantly conditioned by the reactivation of those previous structures that are favourably oriented.

In this regard, here we analyse the relief rejuvenation of an approximately 130 km-long segment of Sierra Morena (Fig. 1b), which is potentially related to recent reactivation of pre-existing structures in the Betics foreland. We focus on the characterisation of the geomorphic signature related to the late Miocene-to-Holocene evolution of Sierra Morena.  and on the discrete tectonic structures that accommodate differential uplift and relief segmentation within the forebulge. We therefore carried out qualitative landform interpretations and quantitative analyses focusing on tributary drainage networks located on

the Guadalquivir River right bank, i.e., those streams draining the Betics forebulge southward. The kinematics of these reactivated structures will be used in this study to explore the recent, even current, tectonic scenario compatible with the observed reactivation pattern. Finally, we will discuss previous geophysical data (Herraiz et al., 2000; De Vicente et al., 2007) and relief modelling (García-Castellanos et al., 2002) as viewed from the new obtained geomorphological and structural data.

Our results prove that the characterisation of relief rejuvenation patterns together with the kinematic analysis of the main relief-

controlling structures provide information about both the roll of reactivated structures in the relief rejuvenation modes and the tectonic setting in which rejuvenation occurs.

## 2 Geological setting and regional topographic features.

In this study, we analyse an approximately 130 km-long segment of the western and central Sierra Morena, including its boundary with the Guadalquivir River valley (Fig. 1). This topographic limit coincides with the contact between two geological

units: the Betic foreland to the north, constituted by the Iberian Massif, and the Betic foreland basin to the south.

The Iberian Massif crops out in western and central Iberia, also constituting the buried basement of the main Cenozoic basins (Fig. 1a). This geological domain belongs to the Variscan Chain, a late Paleozoic orogenic belt, that extends from Morocco to Central Europe. Regarding its distinctive tectonic features, the southernmost Iberian Massif has been divided into three different zones, from SW to NE: the South Portuguese Zone (SPZ), the Ossa-Morena Zone (OMZ) and the Central Iberian

Zone (CIZ). Most of our study area belongs to the SPZ and the OMZ, with the CIZ only represented in its NE limit (Fig. 1b). In our study area, the SPZ is composed of pre-orogenic Devonian to Carboniferous metadetritics, profusely intruded by igneous rocks, which are locally covered by unconformable post-orogenic Permian sequences (e.g., Oliveira, 1990; Simancas, 1983). In its northern limit, a narrow band of slates with interbedded tuffites and quartzites crops out (Matas et al, 2010 b). The OMZ and the CIZ are mainly formed by Neoproterozoic to Carboniferous metadetritic rocks, although lower Cambrian carbonate

formations are well represented. Volcanics are frequently interbedded in the Neoproterozoic rocks of both zones as well as in





the lower Cambrian to Ordovician sequences of the OMZ. Plutonic rocks mainly crop out related to the SPZ/OMZ and ZOM/CIZ boundaries (Matas et al., 2010 a, b).

The current regional limit between the OMZ and the SPZ is located along the transpressive, top to the SW, Southern Iberian Shear Zone (Figs. 1b). It is interpreted as part of a complex, long-lived Variscan suture, including amphibolites of oceanic affinity and high-pressure/low-temperature metamorphic rocks (e.g., Crespo-Blanc and Orozco, 1988; Díaz-Azpiroz and Fernández, 2005; Fonseca and Ribeiro, 1993; Simancas et al., 2003; Pérez Cáceres et al., 2015 and references therein). In our study area, this boundary is hidden to the SE under the infill of the Permian Viar Basin (Figs. 1b, 2a), a NW-SE elongated outcrop, interpreted as a late Variscan intramontane basin with an infill made up of lower Permian detrital and volcanic rocks (Simancas, 1983; García-Navarro and Sierra, 1998). According to previous interpretations, the basin would have been inverted since the end of the Early Permian to the Middle Triassic (García-Navarro and Fernández, 2004). This inversion would have produced both an open N150ºE-oriented syncline and the NW-SE Viar reverse fault system (VFS) along its NE limit. This Permian sedimentary basin coincides with the current Viar river drainage basin (Fig. 1b) downstream of the VFS.

In our study area, the Iberian Massif records a complex Variscan deformation characterised by mainly Devonian to lower Carboniferous, NW-SE trending folds and thrusts (e.g., Expósito et al. 2002; Mantero et al, 2011; Simancas et al., 2003). It should be noted that pervasive fractures of diverse orientation and kinematics have been reported in our study area cutting the main Variscan structures. Many of them have been attributed to the late stages of the Variscan deformation (e.g. García-Navarro and Fernández, 2004), although fractures with Cenozoic activity have been often reported close to the Iberian Massif/Guadalquivir foreland basin limit (e.g., Matas et al., 2010 a; Expósito et al. 2016; Vázquez-Vílchez et al., 2015).

During the Mesozoic tectonic evolution of southern Iberia, the Iberian Massif relief was attenuated towards the SE by the formation of the rifted South Iberian paleomargin, separated from the continental Alboran Domain margin by the Tethys Ocean (Dercourt et al., 1986; Faccenna et al., 2004). The westward migration of the Alboran Domain led to the closure of this Alpine oceanic basin in the Early Miocene, giving rise to the formation of the Betic Chain on the South Iberian paleomargin (Balanyá and García-Dueñas, 1987). In the Middle Miocene, the Betic orogenic load provoked the lithospheric flexure of the Betic foreland (Van der Beek and Cloetingh, 1992), thus creating the Betic foreland basin (i.e., the Guadalquivir foreland basin) coupled with the forebulge just to the north of the Betic mountain front (García-Castellanos et al. 2002). The Guadalquivir foreland basin remained as a marine basin connected to the Atlantic Ocean from the Middle Miocene to the beginning of the Pliocene when the asynchronous westward uplift of the basin led to the coast migration to the SW (e.g., Fernández et al., 1998; Gonzalez-Delgado et al., 2004; Sierro et al., 1996). During the Quaternary, the Guadalquivir River drainage system evolved and the Upper Miocene, shallowing-upward marine sediments were partially covered by alluvial deposits.

In our study area, the oldest infill of the Guadalquivir foreland basin corresponds to Tortonian calcirudites and calcarenites, which unconformably overlie the pre-Mesozoic rocks of the basement. On top of them, Messinian basinal deposits are represented by marly clays. Quaternary deposits consist of alluvial conglomerates, sands, silts and clays related to the Guadalquivir River floodplain and to an extensive strath terrace system (Moral et al., 2013).

Earth **Surface**
**Dynamics**
Discussions

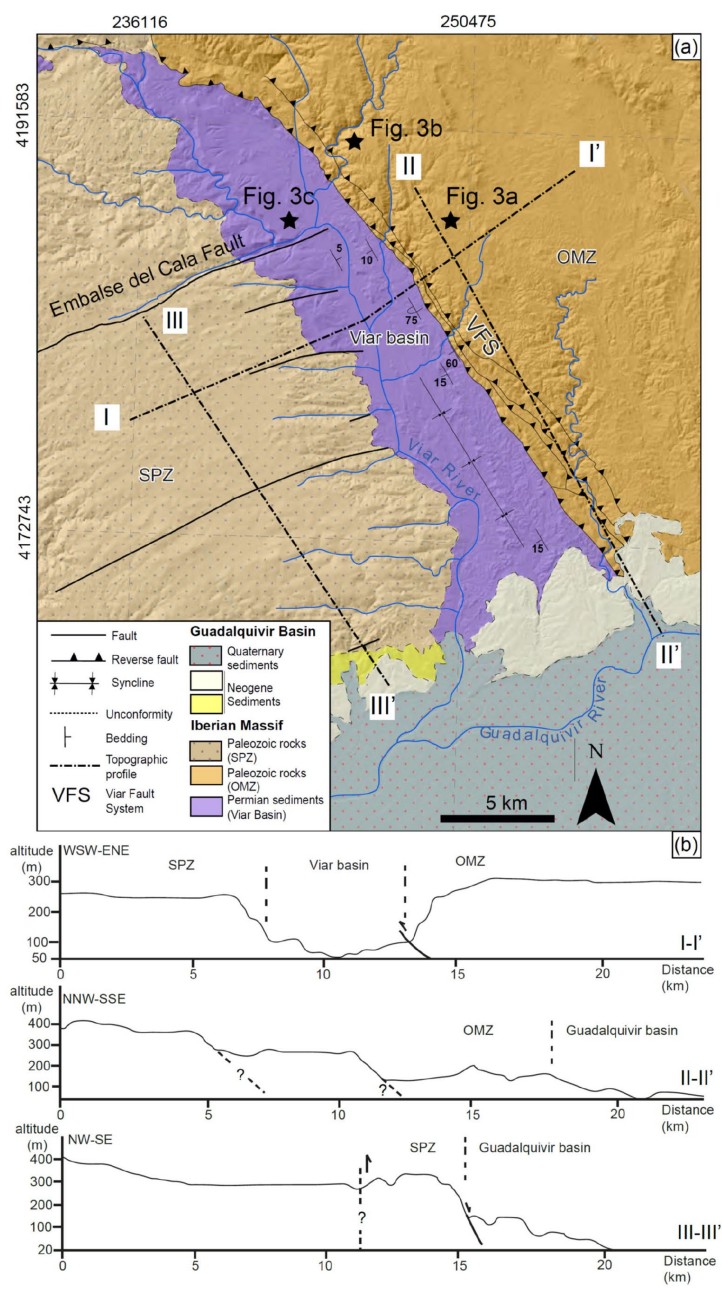


**Fig.2. (a) Simplified geological/hillshade relief map of the Permian Viar Basin (see location on Fig.1b); VFS, Viar fault system; OMZ. Ossa-Morena Zone; SPZ, South Portuguese zone. (b) Topographic profiles across both the Viar basin (I-I´) and the Iberian Massif/Guadalquivir basin boundary (II-II' and III-III'); the dashed line indicates the position of faults inferred from the observed paleo-peneplain downthrow. The altitude base is the 12 m resolution TanDEM-X**
**DEM (Wessel, 2016). The reference system is WGS84 UTM 30N.**



As previously mentioned, the southern boundary of the Iberian Massif basin also separates two distinctive relief domains: Sierra Morena to the north, and the Guadalquivir River alluvial plain to the south, set on the most low-lying areas of the Guadalquivir foreland basin.

The main summits of Sierra Morena are 900 to 1,300 m a.s.l. high, being 900–1,000 m a.s.l. in the study zone. Traverse
topographic profiles across Sierra Morena present a remarkable asymmetry. The relief of the main mountain ridges related to the southern Iberian subplateau and the Guadalquivir valley is about 300–400 m a.s.l. and 800–900 m a.s.l., respectively. Furthermore, the northern boundary of Sierra Morena is a gently dipping transitional piedmont, whereas the southern boundary coincides with a well-defined escarpment of low sinuosity.

The NW-SE orientation of the main streams draining the southern slope of Sierra Morena is determined in their upper reaches
by the Variscan orogenic grain (Fig. 1b). Nevertheless, the Variscan residual relief gradually disappears toward the limit with the Guadalquivir River basin, and is substituted by an erosional, relict surface of plausible Middle-Late Miocene age (Yanes et al., 2019).

At the foot of Sierra Morena, the Guadalquivir River valley is a WSW-ENE-oriented depression limited to the east and south by the Betics mountain front, and to the west by the Atlantic Ocean. In our study area, it is characterised by an overall low-
lying relief with altitudes ranging between 30 and 200 m a.s.l. in the Neogene sediments, and between 10 and 100 m a.s.l. in the Quaternary alluvium. Nevertheless, its easternmost sector exhibits a moderately rugged topography with elevations that reach up to 800 m a.s.l. Furthermore, the Guadalquivir River is in a markedly asymmetric position, located at the northern edge of both its valley floor and alluvial deposits. Terraces that developed on the wider left-hand slope of the Guadalquivir River valley are gently tilted northwards (Moral et al., 2013).

**3 Approach and method**

As stated above, the characterisation of the Sierra Morena relief rejuvenation pattern and the identification of the structures localising such rejuvenation is crucial for our study objective. First, we have searched for signs of relief rejuvenation by applying both geomorphic qualitative methods and geomorphic indexes. The purpose of this was to detect key relief features of relief rejuvenation as well as the potential involved structures (mainly faults). We then proceeded to use field-based
structural analysis methods to determine the kinematic regime of these faults by means of different kinematic indicators such as slickenlines or brittle S-C fabrics.

Our analysis focused on six main drainage basins showing signs of relief rejuvenation. They are located on the right slope of the Guadalquivir river valley, i.e., those draining southern Sierra Morena (Fig. 1b). Additionally, some minor sub-basins have also been included. For this, we used the 12 m resolution TanDEM-X DEM (Wessel, 2016).

In terms of the qualitative approach, we focused our attention on those drainage network features indicative of relief rejuvenation (stream piracy, wind or water gaps, incised meanders, etc.), as well as on the geometry and distribution of a Miocene erosional relict surface that often occupies interfluves. Our qualitative analyses are based on the landscape



morphometry, which has been characterised by means of local relief maps (difference between maximum and minimum elevation by using neighbourhood statistics, 15x15 pixels), as well as slope maps. We also applied several geomorphic indices,

often used in low-rate tectonic regions (e.g., Silva et al., 2003; Pérez-Peña et al., 2009), to better determine the recent tectonic activity associated with the main mountain fronts. After that, we analysed the kinematics of those structures with a position and geometry that are compatible with the observed modes of relief rejuvenation.

Finally, we tested the relationship between those structures associated with the main rejuvenated mountain fronts, the distribution of knickpoints on river profiles in χ-space (Perron and Royden, 2013; Schwanghart et al., 2021) and the drainage

divide network migration (Scherler and Schwanghart, 2020; Schwanghart and Scherler, 2020). This analysis enables us to delve further into the relationship between the analysed faults and the current drainage dynamics. It also sheds light on the relative relief rejuvenation among the drainage basins linked to the fault activity in Sierra Morena.

### 3.1 Geomorphic indexes

We calculated several quantitative geomorphic indices on selected areas of interest, which are very sensitive to changes in the

river equilibrium profile.

- Mountain front sinuosity, *Smf* (Bull and McFadden, 1977; Bull, 1978), was defined as:

$$Smf = \frac{Lmf}{Ls}, \tag{1}$$

where *Lmf* is the length of the mountain foot and *Ls* is the length of the straight line that links the mountain front tips. Values 1 indicate a straight front geometry potentially related to recent tectonic activity of faults. Nevertheless, *Smf* values also depend

on factors such as the lithology and fault dips (Keller and Pinter, 2002; Jiménez-Bonilla et al., 2015). To avoid bias in our results, we applied *Smf* to mountain fronts related to faults dipping between 60º and 80º that bring similar lithologies into contact. Mountain fronts were divided in segments between 2 and 10 km long. The slope break line determining the position of mountain fronts was established at slope values of 8º.

- Valley floor-to-height ratio, *Vf* (Bull and McFadden, 1977), was calculated for streams that run perpendicular to mountain

fronts. *Vf* is defined as:

$$Vf = \frac{Vfw}{(Eld - Esc) - (Erd - Esc)}, \tag{2}$$

where $V_{fw}$ is the width of the valley floor, $E_{ld}$ and $E_{rd}$ are the elevations of the left and right valley divides respectively, and $E_{sc}$ is the elevation of the valley floor.

Two types of valleys are traditionally distinguished according to *Vf* values (e.g., Keller and Pinter, 2002; Silva et al., 2003;

Giaconia et al., 2012; Jiménez-Bonilla et al., 2015): V-shaped valleys (*Vf* < 1) are associated with active down-cutting streams while flat-bottom valleys (*Vf* > 1) testify to large lateral erosion and/or sedimentation related to relatively tectonic quiescence.



- Drainage basin hypsometric curve and hypsometric integral, *HI* (Keller and Pinter, 2002).

The drainage basin hypsometric curve shows the normalised basin area (*a/A*) above a normalised altitude (*h/H*) (Strahler,
1952). The shape of the hypsometric curve has been related to the degree of the dissection of the basin. Upward convex
hypsometric curves are associated with "young" and weakly eroded basins, while concave curves indicate highly eroded basins.
S-shaped curves are frequently associated with moderately eroded basins (Keller and Pinter, 2002), however they can also
feature residual reliefs recently uplifted and reactivated (e.g., Azañón et al., 2012), i.e., relief rejuvenation. This is the reason
why special attention was paid to S-shaped curves. The area below the hypsometric curve, which ranges between 0 and 1, is
the hypsometric integral (*HI*). Values close to 1 indicate weakly eroded regions.

We have plotted the hypsometric curve and calculated HI values for several right tributaries of the Guadalquivir River.

### 3.2 $\chi$-Space analysis of river profiles

- DEM correction and knickpoints extraction.

Pre-processing was performed in order to correct longitudinal profiles prone to data artifacts. Thus, the frequent presence of
artificial dams in Sierra Morena constitutes a source of error that affects the quality of the knickpoint analysis results. To tackle
this issue, we first mapped all the artificial dams as well as the tributary inlets into artificial lakes. The detection of artificial
dams was performed through remote sensing data, such as Google Earth satellite optical images (2018 Landsat Imagery).

The TopoToolbox "mincosthydrocon" function (Schwanghart and Scherler, 2014) was used for the correction with min-max
method. Afterwards, we analysed selected stream networks upstream of fault segments, mostly located at the Sierra Morena
southern mountain front, exhibiting qualitative signals of relief rejuvenation.

On each basin, we extracted knickpoints using the TopoToolbox "knickpointfinder" function (Schwanghart and Scherler, 2014,
2017). The estimate of the uncertainty was performed using the constrained regularised smoothing or CRS technique
(Schwanghart and Scherler, 2017). The tolerance value obtained by the CRS application is 38 m.

- $\chi$ transformation of longitudinal river profiles.

For the river profile analysis, we applied the linear form introduced by Perron and Royden (2013) of the stream power law
(SPL) equation system (Lague, 2014). According to the SPL model, the evolution of the river profile is described as the change
through time *t* in elevation *z* of a channel point located to an *x* upstream horizontal distance, which can be calculated using the
expression:

$$\frac{dz(x,t)}{dt} = U(x,t) - K\,A\,(x)^{\,m}\left|\frac{dz\,(x,t)}{dx}\right|^{n}, \tag{3}$$

where *U* is the uplift, *m* and n are positive an empirical constants and *K* is the erodibility.

Perron and Royden (2013) proposed a transformation of the horizontal spatial coordinates, which linearises the SPL form
(*n=1*), by the integration of *(U/K)^{1/n}* from a base level $x_b$ to an arbitrary upstream point *x* of the channel:



$$z(x) = z(x_b) + \left(\frac{U}{KA_0{}^m}\right)^{\frac{1}{n}} \chi, \tag{4}$$

where $x$ is the horizontal distance of an observation point to the base level $x_b$, $A_0$ is an arbitrary scaling area and $\chi$ is an integration of river horizontal coordinates defined by the equation:

$$\chi = \int_{xb}^{x} \left(\frac{A_0}{A(x')}\right)^{\frac{m}{n}} dx', \tag{5}$$

The advantage of using the linear form of SPL is that those knickpoints, along either the main stem or tributaries, which formed in response to the same fault-related perturbation, will cluster at a specific value of $\chi$, (Perron and Royden, 2013; Schwanghart and Scherler, 2020). Additionally, in steady state conditions, the main stem and tributaries become aligned once the best $m/n$ value has been determined. Thus, transient signals are easier to identify when the transformed profiles deviate from such trend. To find the best $m/n$ value for each basin, i.e., with the best-fit linear relationship between $\chi$ and elevation (Perron and Royden, 2013), we used the TopoToolbox "mnoptimvar" and "robustcov" functions (Schwanghart and Scherler, 2014). In order to apply the $\chi$ transformation, we assume a common base level from which knickpoints propagate. This assumption is suitable for the purpose of the study, since we consider as base level $x_b$ the intersection between streams and fault traces.

- Knickpoint pattern modelling.

In order to cluster the different knickpoint sequences associated with fault activity, we implemented a parametric density distribution technique using the TopoToolbox "rhohat" function (Schwanghart et al., 2021), which also calculates confidence intervals using bootstrapping. In this regard, we estimated the dependence $(\rho)$ of a spatial process (knickpoints) on a spatial covariate ($\chi$ transformed distance from the outlet), implementing the calculus of the parametric density distribution weighted by the knickpoint height. Knickpoint height is the elevation difference between the fitted profile and a knickpoint. The number of bootstrap samples was set to 10000, while the bandwidth of the gaussian kernel was 250 m. 95% bootstrapped confidence intervals were set.

In order to isolate the bias due to the artificial dam's artefacts, we compared $\rho$ values obtained with the entire dataset with those obtained removing knickpoints associated with artificial dams

### 3.3 Drainage divide network analysis

The planform geometry, as well as its relation to topography, has been analysed to investigate possible inconsistencies between expected and observed knickpoint patterns. For that purpose, we followed the method described in Scherler and Schwanghart (2020). In detail, we extracted drainage divides based on the drainage basin boundaries that we obtained for drainage areas at tributary junctions and drainage outlets. We then organised the collection of divide segments into a drainage divide network and ordered it with Topo ordering scheme, in which divide orders increase by one at each junction. As regards topographic metrics, Scherler and Schwanghart (2020) focused on the hillslope relief $HR$, defined as the elevation difference between a point on the divide and the point on the river that it flows to. To quantify the morphologic asymmetry of a divide, the across-



divide difference in hillslope relief (*ΔHR*) is normalised by the across-divide sum in hillslope relief (∑*HR*). The divide
asymmetry index (DAI) was subsequently computed as follows:

$$DAI = \left| \frac{\Delta HR}{\sum HR} \right|, \tag{6}$$

The DAI ranges between 0 for entirely symmetric divides and 1 for the most asymmetric divides.

## 4. Signs of relief rejuvenation in Sierra Morena

### 4.1 Geomorphological features

The relief of the southern half of Sierra Morena is mainly characterised by planar summits alternating with incised valleys.
Summits are regionally oriented in a sub-horizontal attitude or dipping slightly towards the Guadalquivir valley (e.g., Figs. 2b
and 3a), although some exceptions have been found locally. Examples of this anomalous dip have been observed north of
Córdoba (Fig. 4), where the upper edge of the SW-NE Sierra Morena escarpment coincides with the Guadiato eastern divide.
This NW-ward tilting occurs east of a river elbow testifying stream piracy along the Guadiato stream. As a result, the Guadiato
stream bends to NW, thus leaving a N-S oriented wind gap south of the piracy.

These summits often represent remnants of an erosional peneplain which has previously been dated to the Paleogene
(Rodríguez-Vidal and Díaz del Olmo, 1994). Nevertheless, their location adjacent to the Guadalquivir foreland basin suggests
that the relict surface shaping must have been controlled by the local base level of this uplifted marine basin, active during the
Middle-Upper Miocene (Yanes et al., 2019). Locally, summits are occupied by patches of Neogene shallow marine sediments
belonging to the Guadalquivir foreland basin. They unconformably overlie the pre-Mesozoic basement and are uplifted several
tens, or even hundreds, of metres above their equivalent in the Guadalquivir River valley (e.g., Figs. 1b and 4a, d). Irrespective
of their nature, flat summits often describe a roughly N-S stepped relief segmentation i.e., parallel to the Sierra Morena southern
boundary.

Additionally, there is significant WSW-ENE segmentation of the Miocene relict surface across the Permian Viar Basin, which
partly coincides with the Viar River lower drainage basin (Fig. 2a). This NW-SE depression is open to the Guadalquivir River
valley. The average elevation of its low-lying relief is similar to that of the hills shaped on the Miocene and Plio-Quaternary
rocks of the northern Guadalquivir valley. The Permian Viar Basin is flanked by NW-SE-oriented ranges, markedly incised
by tributaries of both the right and left banks of the Viar River. The resulting summits have smooth slopes that rarely exceed
5º towards the SSE (Fig. 2b). In contrast, their boundaries with the Permian Viar Basin are abrupt with averages slopes of 45%
on its eastern boundary, coinciding with the VFS trace, and 20% on its western boundary. When compared on both sides of
the VFS trace, the erosional relict surface located on the OMZ summits (i.e., the hanging wall) is at a higher topographic level
(up to 100 m higher) than on the SPZ summits (i.e., VFS footwall).





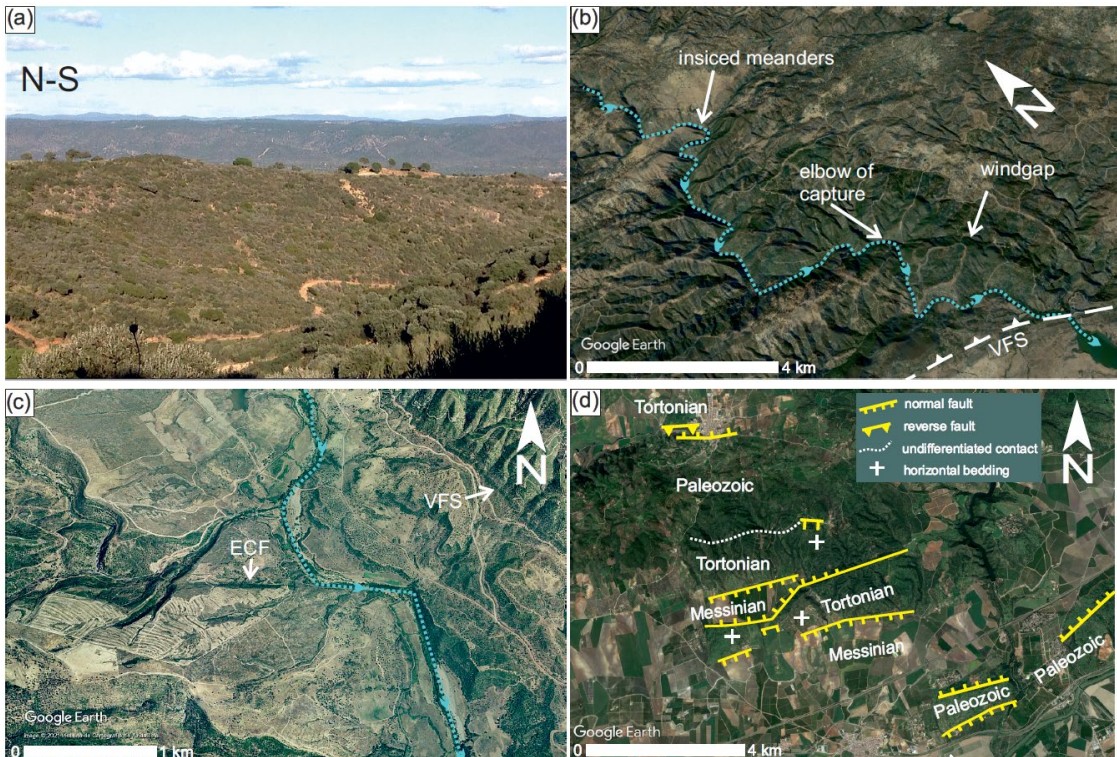

**Fig. 3. (a) Relict peneplain on top of the Ossa Morena Zone (background), just east of the Viar basin; the photograph is taken from the South Portuguese Zone (foreground). (b) 2019©Google Earth oblique view showing a stream elbow testifying to a piracy along the Viar River at its intersection with the Viar fault system, VFS. (c) Google Earth view of the Viar River deflection associated with the Embalse del Cala fault, ECF; image2022©Instituto de Cartografía de Andalucía from 2004, before the current dam construction. See location of (a), (b) and (c) on Fig. 2a. (d) Faults related to the NNW-SSE relief segmentation defining horsts and grabens in the southernmost Sierra Morena (see location on Fig.1b); faults are drawn on a 2019©Google Earth image.**

Drainage networks follow a mainly dendritic pattern, paired with local parallel networks. The main streams trend NW-SE in their upper reaches —following the Variscan orogenic grain— and bend N-S close to the boundary with the Guadalquivir valley (Fig. 1b). This is conditioned by the southward increasing topographic gradient between the Iberian Massif and the Guadalquivir basin. Secondary stream orientations, however, are often controlled by fractures, particularly in the SPZ, where lower order streams of the Viar River follow the trend of SW-NE to W-E fractures (Fig. 2a).

Streams often display features associated with relief rejuvenation such as incised meanders (e.g., Fig. 3b), perched terraces or stream elbows due to both river capture (e.g., Fig. 4a) and fault-related stream deflections (e.g., Fig. 3c). Long profiles of both main and secondary streams often show typical non-equilibrium, upwards-convex geometry (Fig. 8) and display knickpoints that are not related to lithological contrast. Long profile analysis will be carried out below in section 6.



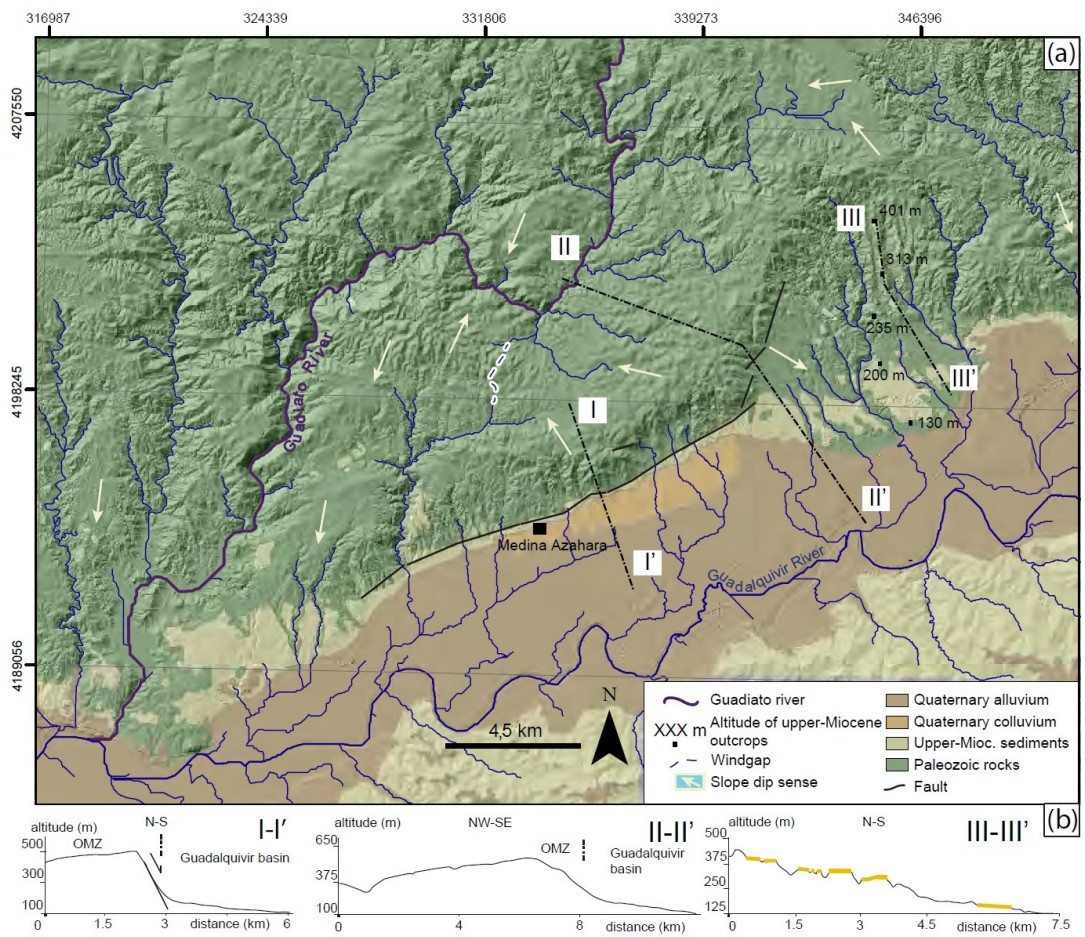

**Fig.4. (a) Simplified geological/hillshade relief map of the Guadiato River area (S6); see location on Fig.1b. (b)**
**Topographic profiles across the Iberian Massif/Guadalquivir basin boundary. The altitude base is the 12 m resolution**
**TanDEM-X DEM (Wessel, 2016). The reference system is WGS84 UTM 30N.**

### 4.2 Geomorphic indexes

A quantitative analysis (Fig. 5) was performed on six drainage basins belonging to the main right tributary streams of the
Guadalquivir River (S1 to S6 in Fig. 1b). From west to east, these basins belong to the following rivers: Rivera de Huelva-
Cala (S1), Viar (S2), Guadalbarcar (S3), Retortillo (S4), Bembézar (S5) and Guadiato (S6). Most of these rivers are particularly
incised upstream of the Sierra Morena southern mountain front. An exception is the Viar River (S2), where the incision occurs
upstream of the VFS. Additionally, some second-order tributaries within S2 have also been analysed by means of geomorphic
indexes.



Overall, those mountain fronts related to the southern Sierra Morena escarpment yield low *Smf* values, thus corresponding to
straight mountain fronts (Table 1). In detail, those values obtained for segments of the WSW-ENE Sierra Morena escarpment
vary from 1.15 to 1.3, pointing to tectonically active mountain fronts (class 1 of Silva et al., 2003). In the same way, *Smf*
related to the Permian Viar Basin's eastern boundary, i.e., coinciding with the VFS, are between 1.2 and 1.4. Values obtained
for the western boundary are slightly higher, between 1.2 and 2.0.

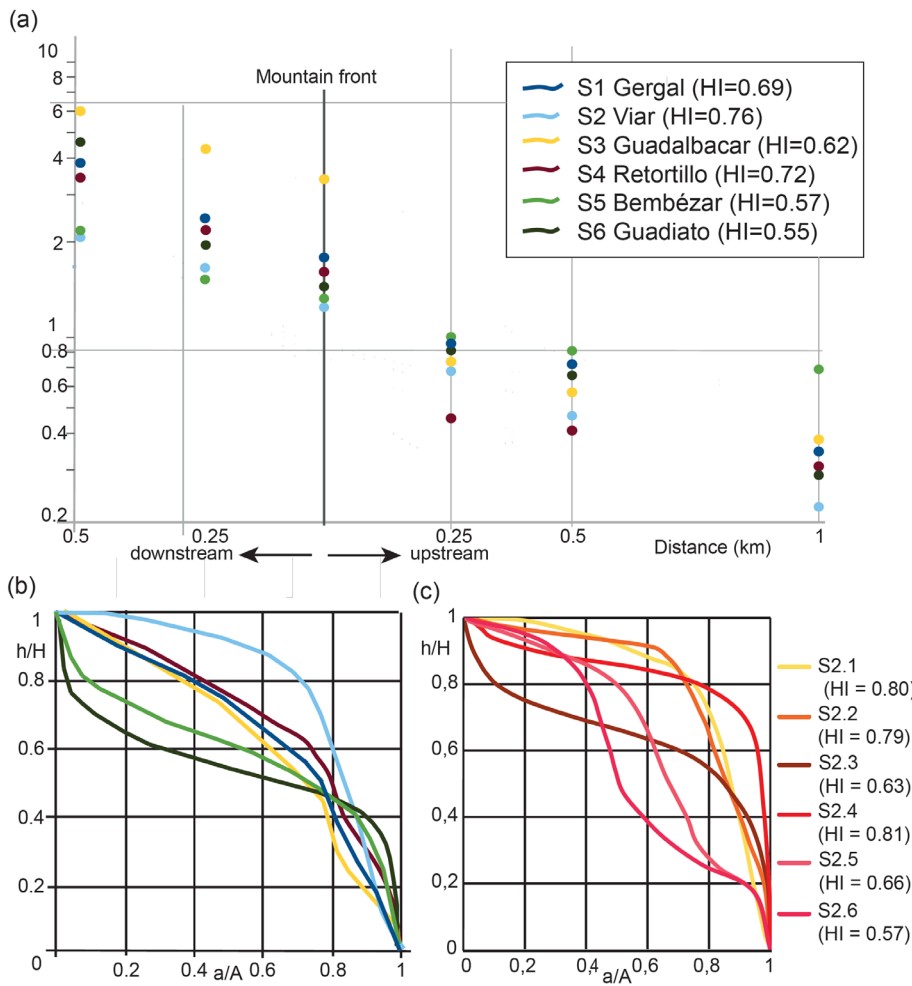


**Fig.5. Geomorphic indexes. (a) Vf values for the main streams of S1 to S6, upstream and downstream of fault-related**
**mountain fronts (see Fig.7 for fault location). (b) Hypsometric curves and hypsometric integral values, HI, for the same**
**streams as in (a). (c) Hypsometric curves and HI for sub-basins on the right slope of the Viar river valley. The altitude**
**base is the 12 m resolution TanDEM-X DEM (Wessel, 2016). The reference system is WGS84 UTM 30N.**




Although the considered mountain fronts are crossed by the lower reaches of the analysed streams, *Vf* values are less than 1 (Table 1 and Fig. 5a), therefore associated with V-shaped valleys, when measured upstream from both the Sierra Morena escarpment and the VFS. Conversely, these values are systematically greater than 1 downstream from the mountain fronts, thus indicating the relationship between river incision and these landforms. On the other hand, there is little variation in the *Vf*

values that each stream yields upstream from the mountain fronts.

Hypsometric curves for S1 to S6 main streams display dominant upward convex geometries (Fig. 6b), with HI values > 0.5 (Table 1 and Fig. 5b), suggesting, in principle, relief rejuvenation of their watershed. S5 and S6 curves are slightly S-shaped with relatively high *h/H* values in its middle course, also showing the lowest *HI* values. This may indicate that relief rejuvenation is localised in its lower course.

Similar hypsometric curves geometries and *HI* values have been obtained for tributaries of the Viar River (Fig. 5c), indicating that the SW boundary of the Permian Viar Basin has also undergone relief rejuvenation.

**Table 1. Geomorphic indexes for the six main drainages basins (S1 to S6)**

|      | S1   | S2   | S3   | S4   | S5   | S6   |
|------|------|------|------|------|------|------|
| Smf  | 1,32 | 1,25 | 1,18 | 1,21 | 1,15 | 1,2  |
| Vf   | 0,82 | 0,79 | 0,68 | 0,43 | 0,88 | 0,85 |
| HI   | 0,69 | 0,76 | 0,62 | 0,72 | 0,57 | 0,55 |

**5 Structures related to relief segmentation and rejuvenation**

As described in previous sections, the topographic distribution of Sierra Morena is regionally defined by a NNW-SSE segmentation that is primarily localised in the Sierra Morena southern slope break, but that also extends northwards, often conditioning the orientation of the secondary stream networks (e.g., Fig. 2a). Thus, this segmentation is defined not only by the topographic step at the southern limit of Sierra Morena, but also by the tilting and displacement, to the north, of both the

Miocene erosional paleo-surface and Miocene marine deposits (e.g., Fig. 2b and 4b, d).

We have found that this NNW-SSE segmentation is spatially related to faults that trend from SW-NE to WNW-ESE, the former being the best represented (Fig. 6). Most of them exhibit steep surfaces dipping with either N or S component. Their slickensides indicate slip senses that vary from dominant normal dip slip to dominant right-lateral slip, with the oblique, right-lateral normal faults being well represented. Although the normal slip component is dominant, reverse component also occurs.






Earth **Surface**
Dynamics
Discussions

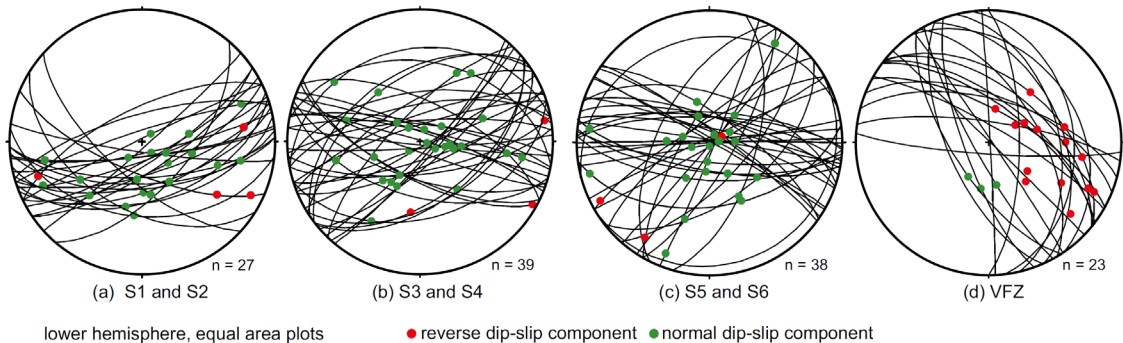

**Fig. 6. Stereoplots of faults related to relief rejuvenation. (a), (b) and (c) faults associated with NNW-SSE relief segmentation. (d) Fault surfaces measured in the VFS. Plots have been made with Faultkin 8 by Allmendinger (2019, www.geo.cornell.edu/geology/faculty/RWA/programs/faultkin.html).**


Some of these faults involving Paleozoic rocks seem to have formed by reactivation of previous structures. For example, in igneous rock outcrops of both the SPZ and the Permian Viar Basin, these faults are evenly distributed with a spacing of metric order, thus pointingto the reactivation of previous joints. In the same way, slickensides indicating normal to dextral faults have been observed along the WSW-ENE Embalse del Cala fault (figs. 2a, 3c), a Variscan, left-lateral structure located west of the

Permian Viar Basin (García-Navarro and Férnandez, 2004).

As mentioned above, this fault system is particularly developed in the Sierra Morena escarpment, defining its southern mountain front. The presence of both relict surface remnants and shallow water Miocene rocks along the top of the escarpment enables a minimum vertical throw of the Paleozoic basement to be established. This vertical throw seems to have been greater in the east of our study area (up to 480 m north of Córdoba, in the south of S6; Fig 4b, c and d), gradually decreasing toward

the west (≈ 50 m west of the Permian Viar Basin, in the south of S1; Fig. 2a). In the eastern sector, the vertical throw is clearly exhibited by the abrupt W-E to SW-NE escarpment located to the NW of Córdoba. At its foot, escarpment-derived colluvium of debris-facies can be observed, both pre- and post-dating the Medina Azahara archaeological site (Fig. 4a), which was abandoned and vandalised soon after 1036 A.D., (López-Cuervo, 1983). In the northernmost archaeological site, the most recent colluvium (ca. 2 m thick) seals part of the ancient buildings and incorporates scarce fragments of material from Medina

Azahara buildings, such as ceramic tiles and wall bricks.

In addition to this regional NNW-SSE relief segmentation, a significant WSW-ENE relief segmentation also occurs across the Permian Viar Basin, i.e., roughly parallel to the overall Sierra Morena escarpment trace. In the NE limit of the Permian Viar Basin, this segmentation is markedly localised along the VFS which consists of mainly NW-SE trending faults. They present moderate to steep NE-ward dips and their slickensides indicate reverse left-lateral slip.

In contrast to the NE boundary of the Permian Viar Basin, relief rejuvenation in its SW boundary does not seem to be conditioned by NW-SE-oriented faults, but rather is significantly distributed in diverse structures. Here, the SPZ rocks are





unconformably overlain by gently NE-ward, Permian rocks (SW limb of the Viar synform), and signs of relief rejuvenation seem to be mostly related to the WSW-ENE fault system. The different relief rejuvenation modes observed to the SW and NE of the Permian Viar Basin explain that, although *HI* yield similar values for both boundaries, *Smf* values are higher in the SW

(i.e., the SPZ/Viar basin boundary). Here, differential uplift does not seem to be accommodated by a discrete structure such as the VFS, but rather mainly accommodated by distributed structures of limited extent.

## 6 Knickpoint pattern analysis

The geomorphological and structural data set up in previous sections suggest that the relief rejuvenation and segmentation observed in our studied area is related to two main fault systems that operated coupled with the regional flexural uplift of the

forebulge area. The first one consists mainly of WSW-ENE-oriented faults, highly localised along the similarly oriented southern Sierra Morena mountain front. The second one is the NW-SE-oriented VFS, which limited the Permian Viar Basin

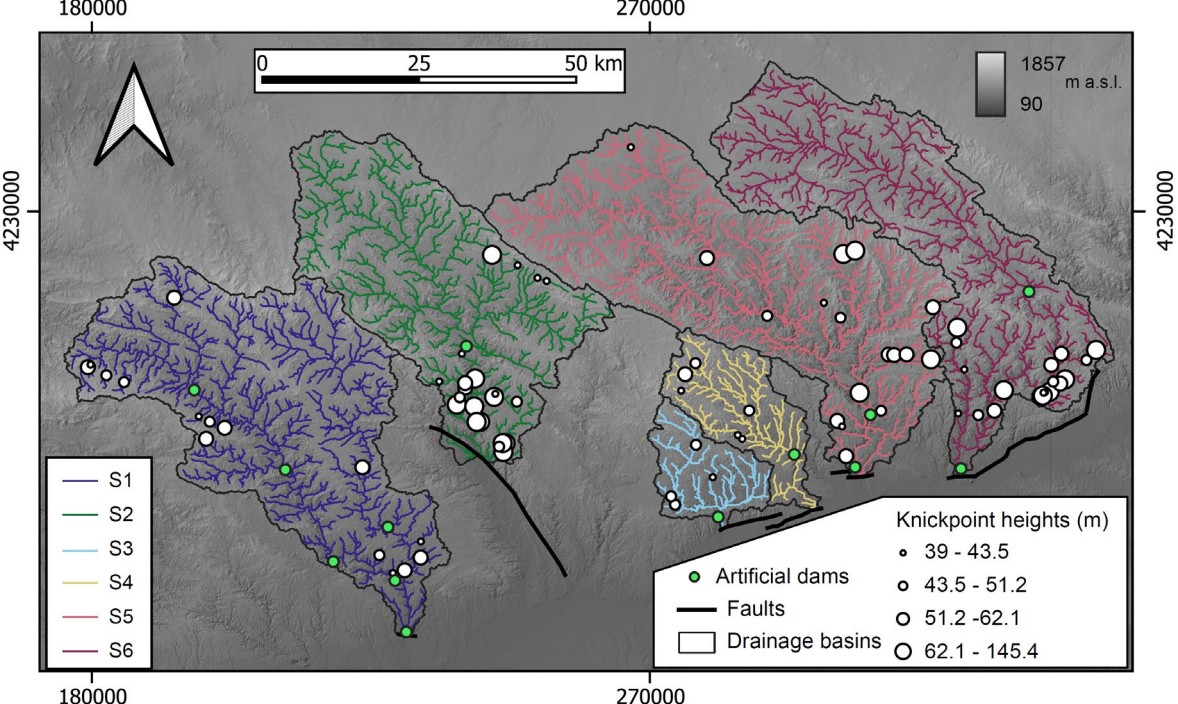

**Fig.7. Stream networks upstream of fault-related mountain fronts. Knickpoints sized by their height as well as artificial dams are also reported. The elevation base is the 12 m resolution TanDEM-X DEM (Wessel, 2016). The reference system is WGS84 UTM 30N.**





to the NE (Fig. 1b and Fig. 2a). All the analysed basins are crossed in their lower reaches by one of these fault-bounded
mountain fronts (Fig. 7), either related to the Sierra Morena slope rupture (S1, S3, S4, S5 and S6) or to the VFS (S2). Thus,
the drainage network located upstream of faults may have recorded their activity through knickpoint migration.

In this regard, the distribution of knickpoints upstream of the fault-related main mountain fronts is analysed in this section and
the distribution is compared with that predicted by our knickpoint pattern models (Schwanghart et al., 2021). This analysis
will enable the characterisation and comparison of recent tectonic activity along different sectors of Sierra Morena, as well as
enabling the deviation between observed and predicted knickpoints to be explored in terms of divide network re-organisation.
To perform $\chi$ transformations of longitudinal river profiles, we have calculated for each basin the $m/n$ ratio in Eq. (5) that
minimises the variability of elevation values for similar values of $\chi$ (Table 2):

**Table 2.** *m/n* values for basins S1 to S6.

| S1 | S2 | S3 | S4 | S5 | S7 |
|---------|---------|---------|---------|---------|---------|
| -0.4061 | -0.4356 | -0.2985 | -0.1679 | -0.4989 | -0.4131 |

Figure 8 shows the position on both main stream and tributary longitudinal profiles for basins S1 to S6. Horizontal coordinates
are displayed in both regular distance (left side) and $\chi$ space (right side). Coordinates 0,0 correspond to the intersection between
the main stream and the fault-related mountain front. Artificial dams and tributary inlets into reservoirs are also projected along
profiles. Finally, covariance estimates ($\rho$) for knickpoint patterns on a spatial covariate ($\chi$ transformed distance from the outlet)
are reported with and without artificial dams.

The longitudinal profiles of main streams, except for S4, are partially conditioned by the presence of dams, with this being of
particular note in S1, which exhibits five dams distributed over 100 km. Despite these dam-related interruptions along profiles,
the S3 and S4 plots are seen to display a concave-up shape, whereas those corresponding to S2 and S6 show a conspicuous
convex-up geometry.

The knickpoint distribution along profiles presents certain differences from one drainage basin to another. In S2, S5 and S6,
most of the knickpoints are related to the middle and lower values of both $x$ (Fig. 8c, i, k) and $\chi$ (Figs. 8d, j and l). Conversely,
knickpoints are evenly distributed along S1 plots (Fig. 8a, b). Finally, the scarce knickpoints of S3 and S4 are located in middle
and high values of the horizontal coordinates (Fig. 8e, f, g, h).

The $\chi$ space profiles offer a better perspective for discussing knickpoint pattern dependence. Indeed, the covariance estimate
(both with and without artificial dams) highlights how the majority of the higher knickpoints are in the lower part of S1, S2,
S5 and S6, whereas in the upper part, several minor knickpoints form secondary peaks in the covariance trend. Knickpoints
are particularly clustered in S2, where most of them are distributed in a very narrow strip corresponding to the lower part of
the drainage basin, with $\chi$ values around 1,000–1,500 m, and elevations around 200 m above the outlet point. Furthermore, S6



knickpoints are clearly grouped in the lower part of the basin, though to a lesser extent than in S2, with $\chi$ values between 2,000–3,500 m and elevations between 200 m and 400 above the outlet point.










**Fig. 8. S1-S6 stream network plots. (a), (c), (e), (g), (i) and (k) river longitudinal profiles; (b), (d), (f), (h), (j), and (l) χ transformed longitudinal profiles. Knickpoints, coloured by elevation and sized by height, artificial dams, and tributary**

**inlets into the reservoirs are projected on profiles. The covariance estimates for point patterns, calculated on the χ transformed distance from the outlet (spatial covariate), are reported with and without dams.**



From the comparison of the covariance computed with and without artificial dams in the dataset, it can be observed that the
dependence of the knickpoint pattern is variably affected by the occurrence of artificial dams. In this regard, such dependence
is strong in S1, S3, S5 and S6, as can be inferred by the shifting of the intensity peaks when covariate values obtained with
and without dams are compared. However, the dependence of the knickpoint pattern to artificial dams does not seem to be
significant in S2, only raising slightly the intensity peak of the plot. S4 is not at all biased, since no knickpoints were
automatically extracted that coincide with the only artificial dam of this drainage basin. In this regard, the plano-altimetric
configuration of some dams cannot exclude the pre-existence of a knickpoint favouring the dam location. Examples of this are
the artificial dam of S2 or the highest one of S6, whereby the location on the $\chi$-plot fits with the values range of knickpoints.
As can be observed in Fig. 8, only S2 and S6 networks show a strong consistency between the model and the location of a
robust number of actual knickpoints. Both S3 and S4 show peaks of density values above the average. However, there is
considerable uncertainty surrounding these results because of the few knickpoints located in the peaks of $\chi$ value. Instead, the
intensity peaks of S1 and S5 are highly distributed and lower than the average. In detail, S1 shows values slightly higher both
in the lower and in the upper parts of the basins, while S5 is affected by greater uncertainty, not showing any outstanding
density peak of expected knickpoints. The notable difference between the actual and expected knickpoint patterns, especially
in S1 and S5, will be analysed in the discussion section.

## 7 Discussion

### 7.1 Geomorphic response to tectonic uplift

As described in section 4, the Pliocene and onwards relief rejuvenation in our study area may be inferred from both the
interpretation of observed geomorphological features and the results yielded by the applied quantitative methods. In addition,
our structural results suggest that the distribution of such rejuvenated relief is greatly controlled by the position and orientation
of two main fault systems with Quaternary activity.
The activity of overall WSW-ENE faults has strongly contributed to the roughly NNW-SSE relief segmentation of the foreland,
which is particularly localised along the topographic escarpment separating Sierra Morena from the Guadalquivir foreland
basin. The existence of both  relict surface remnants and Neogene outcrops along the top of the escarpment has allowed us to
compare the relative uplift accommodated along this escarpment and its variations from east to west.
In this regard, the foreland escarpment north of Córdoba is particularly high and exhibits a higher topographic gradient in
comparison to other sectors located to the west. In this eastern sector, faulting also appears to be related to the recent NW-
ward tilting of the foreland relict surface in the Guadiato River lower course (S6), assumed to be originally dipping towards
the foreland basin. This uplift-related tilting explains the sharp stream elbow testifying to the piracy of the Guadiato River
whereby the abandoned lowermost course has remained as a wind gap.
As stated previously, well developed, debris-facies colluvium is deposited at the foot of this escarpment post-dating the ancient
city of Medina Azahara. Thus, this debris accumulation points to the instability of the escarpment (e.g., Forman et al., 1991)



over the last 1,000 years. The thickness of such colluvium is similar to that reported for recent, even active, faults (e.g., Koukouvelas et al. 2005). This recent activity is in line with some historical earthquakes recorded in Córdoba at the end of the 10th century (www.iagpds.ugr.es: Andalusian Institute of Geophysics and Seismic Disaster Prevention Centre). In this regard, a recent archaeological and seismological study on Medina Azahara describes oriented damages, compatible with NW-SE-
oriented ground motion, which could be reasonably related to the destruction of the city at the beginning of the 11th century (Rodríguez Pascua et al., 2021).

Towards the west, the N-S segmentation of southern Sierra Morena is still observed; however, the topographic escarpment becomes lower and less sharp. In contrast, relief rejuvenation seems to be particularly localised along the NW-SE oriented Permian Viar Basin limits, particularly along the VFS, where the fault trace coincides with the abrupt escarpment separating
the Permian Viar Basin and the OMZ.

This outstanding WSW-ENE topographic segmentation across the Permian Viar Basin is also characterised by the topographic discontinuity of the erosional relict surface, which is located at higher altitude on the NE side of the basin (the hanging wall of the VFS; Fig. 2b). Given the Miocene age attributed to such relict surface, this discontinuity is in agreement with a Pliocene or Quaternary activity of the VFS, which uplifted the OMZ. Furthermore, the distribution of the Neogene sediments along the
SE limit of the Permian Viar Basin, together with the lack of said sediments in the basin, point to the same conclusion. Thus, whereas the floor of the Viar drainage basin is currently at the same altitude as the northernmost Neogene outcrops of the Guadalquivir basin, these sediments do not extend into the Viar drainage basin. This may indicate that the uplift of the forebulge region (i.e. Sierra Morena), likely accommodated in part by the WSW-ENE fault system, was prior to the Viar River drainage basin downthrow, suggesting that the VFS reactivation is relatively young within the relief rejuvenation event in
Sierra Morena.

Thus, although there is qualitative and quantitative evidence of recent tectonics in the whole study area, the activity seems to be particularly manifest in sectors where basins S2 and S6 developed. The same conclusion is suggested by the comparison between the χ-plots obtained for basins S1 to S6. In this regard, S2 and S6 show a greater consistency between the non-parametric dependence model and the current positions of knickpoints. In these basins, most of the knickpoints cluster in low
χ-values, thus pointing to the recent propagation of the perturbation from its source, i.e., faults located in the escarpments of Sierra Morena (S6) and the VFS (S2).

In Fig. 9, the elevation of knickpoints and dams has been plotted in $\chi$ horizontal coordinates, and the best linear fits and related regression coefficients have been calculated for basins S1 to S6. According to Eq. (4), the slope of $\chi$-profiles is directly proportional to the uplift rate ($U$) and inversely proportional to the erodibility ($K$). Since there are no strong lithological
contrasts in the study area, the slope of plots can be interpreted as a proxy of the tectonic activity. Assuming this, S2 and S6 show the greatest uplift together with S5. The S1 plot shows a moderately high slope that could also be related to recent tectonic activity. It is worth noting that the S2 and S6 plot slopes would become steeper if those knickpoints in higher $\chi$-values, disconnected from the main clusters, were removed. In addition, the quality of their regression coefficients would increase given that the scarce knickpoints in the upper part of the plot hamper an optimal linear interpolation. In any case, the slopes of



plots for S2, S5 and S6 suggest that these basins are located in sectors with the greatest and, at least for S2 and S6, most recent uplift within our study area.

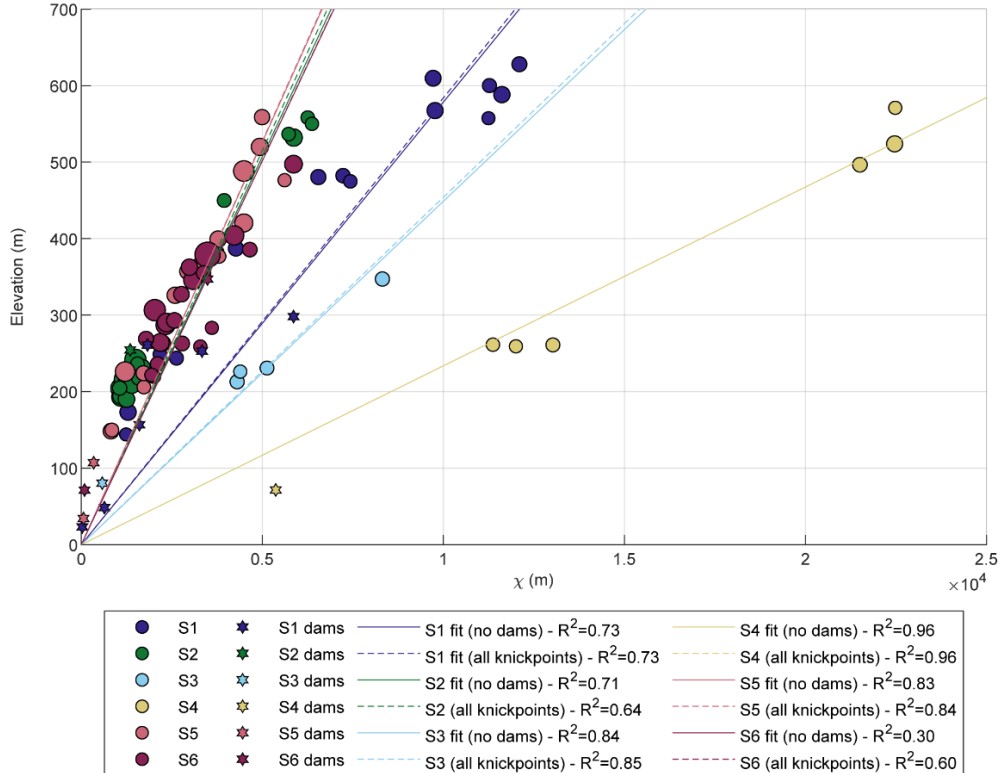

**Fig. 9. χ versus elevation plot of knickpoint and artificial dams related to each drainage basin. The linear fits and the associated regression coefficients with and without artificial dams are reported.**

Although S2, S5, S6 and, to a lesser extent, S1 show significant plot slopes, potentially related to recent uplift, only S2 and S6 are characterised by strong consistency between observed and predicted knickpoints. Conversely, S1 and S5 deviated from the

obtained model. The reason for this deviation could lie in processes related to divide network reorganisation, which have been proven to alter the upstream propagation of knickpoints (Schwanghart and Scherler, 2020). In this regard, the DAI metric sized by divide order (Fig. 10) reveals significant drainage reorganisation and enables the divide migration direction to be inferred. Such reorganisation alters drainage areas and discharge, and thus impacts on knickpoint celerities, which in return will result in more scattered knickpoint locations (Schwanghart and Scherler, 2020). Specifically, S1 is affected by a strong internal

divide migration towards the east, suggesting that the western reach of S1 may have captured the eastern reach. Moreover, the

Earth **Surface**
**Dynamics**
Discussions

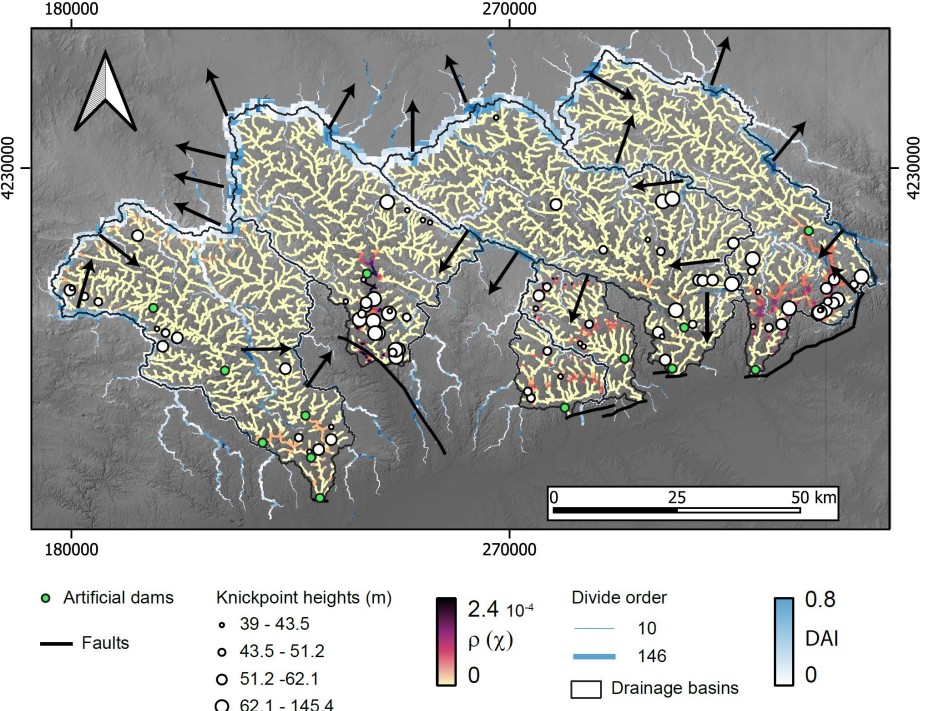

**Fig.10. Observed and expected spatial patterns of knickpoints in the selected Sierra Morena basins. The intensity (ρ) values are computed on the knickpoint dataset without artificial dams. The drainage divide network asymmetry (DAI) sized by order is reported and the inferred movement is indicated by the black arrows. The elevation base is the 12 m resolution TanDEM-X DEM (Wessel, 2016). The reference system is WGS84 UTM 30N.**

main basin divide is also moving, decreasing the basin drainage area, likely affecting the locations of the upper reach knickpoints. The SE sector of the divide between S2 and S5 is moving SW-ward, thus decreasing the drainage area of the former, whereas the NW sector of the S2 divide is moving towards NW, increasing its drainage area. The large uncertainties around the single knickpoints occurring higher up in the drainage basin can be explained by such drainage reorganisation. Nevertheless, S2 is experiencing a great stability that explains the strong consistency between the observed and expected knickpoint patterns in the lower part of the basin. S5 divides seem to be migrating SW-wards, gaining area from S2 and S4, and losing area in the lower part of the drainage basin along the S5-S6 main divide and, also, along minor divides. Nevertheless, in the upper part of the same divide, NE-ward migration is occurring, which, combined with the aforementioned SW-ward migration of the lower basin, gives rise to a clockwise rotation of the divide. Such movement will likely cause S6 to lose area at the higher part of the basin. This divide reorganisation could account for the difference in consistency among the model and the current knickpoint locations in S5 and S6. This reorganisation allows S6 to preserve a robust cluster of knickpoints in its





lower part. Conversely, the uncertainty and lower-than-average intensity peaks shown by $\chi$ space profile of S5 (Fig. 8j) can be explained by such drainage reorganisation.

## 7.2 Matching the geomorphic results with the tectonic scenario

The Sierra Morena is characterised by significant mechanical discontinuities achieved during its complex tectonic evolution. These discontinuities are predominantly defined by both NW-SE-oriented Variscan structures, and mainly WSW-ENE faults related to the Alpine orogenic cycle, which eventually gave rise to the Betic chain building during Neogene times. Such orientations coincide with those found for faults with Quaternary activity, thus suggesting that the observed relief rejuvenation pattern is strongly determined by the reactivation of previous structures that are accommodating the current strain partitioning. The overall WSW-ENE trend of faults associated with the Sierra Morena southern escarpment, and in general to its NNW-SSE relief segmentation, is in agreement with those expected to accommodate extension by flexural bending in the Betics forebulge, i.e., parallel to this topographic high. In this regards, normal faults with a dominant dip-slip component, therefore producing extension roughly orthogonal to the forebulge trend, are frequent. Nevertheless, the overall cartographic WSW-ENE trend of the escarpment is at smaller scale defined by faults whose orientations range from SW-NE to WNW-ESE. This suggests that not only newly formed, forebulge-related faults may be involved, but also those Variscan structures with an orientation that favoured their reactivation. Examples of Variscan WNW-ESE structures recently reactivated as normal faults have been previously described in the Sierra Albarrana area (SA in Fig. 1c; Herraiz et al., 1996). Additionally, reactivated late Variscan SW-NE to WSW-ENE fractures seem to be also present. Examples include joints exhibiting striated surfaces in late Paleozoic igneous rocks, or the Embalse del Cala Fault (figs. 2a, 3c), a left-lateral fault (García-Navarro and Fernández, 2004) interpreted in this work as a reactivated normal dextral fault.

Although most of these structures have a normal slip component, many of them are roughly vertical or even steeply dipping reverse faults, suggesting that current vertical extrusion is significant in the forebulge zone. This may indicate that compression is tightening this flexural relief, also explaining the reactivation of the Variscan NW-SE-oriented VFS as a shortening structure. This tightening is consistent with previous seismotectonic analysis in Sierra Morena, showing that focal mechanisms of reverse faults are deeper than those of normal faults (Herraiz et al, 1996). This vertical distribution of the stress matches with tangential longitudinal strain by lithospheric buckling. In this respect, previous numerical models of the paired Betic foreland basin-forebulge (i.e., Guadalquivir basin/Sierra Morena) evolution have settled that the current altitude of Sierra Morena cannot be explained only by lithospheric flexure of the South Iberian basement. Therefore, these models invoke compressional tectonic forces to enhance the Sierra Morena uplift by lithospheric folding (García-Castellanos et al., 2002; Cloething et al, 2002). This is in line with the amplitude/width ratio of Sierra Morena, which is significantly higher than other forebulge examples (e.g., Niviere et al., 2013).

This compressional setting has been often contextualised within the overall NW-SE-oriented compression of the Iberian Peninsula, linked to the convergence between the Eurasian and the African plates (e.g., Herraiz et al., 2000 and references therein). Nevertheless, this NW-SE-oriented compression hardly explains either the dextral component that is frequently





exhibited by the WSW-ENE fault system or the reverse slip accommodated by the VFS, which is oriented parallel to this compression direction. In this respect, our results seem to be more compatible with forebulge compression due to the intraplate propagation of the Betics transpressional deformation front. Indeed, the Neogene to recent dextral transpressional kinematics recorded in segments of the western Betics is consistent with a N99°E–N109°E trending horizontal velocity vector (Díaz-Azpiroz et al., 2014, Barcos et al., 2015, 2016) attributed to the west-ward migration of the western Gibraltar Arc (Balanyá et al. 2012, Gutscher et al., 2002). This shortening orientation would have reactivated the VFS as a left-lateral reverse fault. Additionally, the dextral transpressional deformation of the forebulge would generate SW-NE-oriented extension. This extension would imprint an overall dextral kinematics on the WSW-ENE to WNW-ESE faults, which would be characterised by a high strain partitioning due to the different orientation of the reactivated structures. Interestingly, fault population analysis techniques applied to sectors nearby our study area have yielded similar SW-NE-oriented extension ($S_{min}$ trend, Herraiz et al., 2000).

## 8. Summary and conclusion

Our results show that the combination of geomorphological and structural analyses of ancient areas can be a powerful tool not only to explore the roll of reactivated structures on relief rejuvenation, but also to delve into the tectonic setting responsible for such reactivation. This combined analysis can be exported to any long-lived region in order to compare the proposed tectonic setting with the observed geomorphological features. In our case, we have interpreted in tectonic terms the main geomorphological traits of Sierra Morena, a flexural relief (i.e., the forebulge) generated in the Betics foreland due to the load exerted by the orogenic pile.

The analysed drainage network of Sierra Morena shows a wide range of signs of relief rejuvenation such us incised meanders, fault-related streams deflection and stream piracy. Additionally, the interfluves are often constituted by either patches of Miocene rocks belonging to the Guadalquivir foreland basin or remnants of a peneplain shaped at the foreland basin base level. These two stratigraphic and geomorphic markers are not only uplifted above the foreland basin, but are also commonly tilted and in a stepped altimetric disposition.

The qualitative geomorphological characterisation, together with the use of geomorphic indexes, evidence the relationship between mountain fronts and Quaternary activity of both newly formed and re-activated faults. The main faults controlling relief rejuvenation belong to two different groups: the first is mainly composed of WSW-ENE, right-lateral normal faults, responsible for the NNW-SSE relief segmentation of Sierra Morena as well as the abrupt escarpment between this range and the Guadalquivir foreland basin; the other fault group is localised on the Viar fault system (VFS), previously interpreted as a late-Variscan, left-lateral reverse fault system, and is responsible for WSW-ENE relief segmentation in the western sector of the study area.

Knickpoint pattern modelling, based on $\chi$-space and performed on six drainage basins, enables us to define the relationship between the activity of the analysed structures and the current geomorphological features of Sierra Morena in a more precise

Earth **Surface**
**Dynamics**
Discussions

EGU

manner. The main stems of these selected basins are crosscut by fault segments, which are related to either the Sierra Morena southern escarpment or the reactivated VFS. The planimetric and altimetric knickpoints clustering on $\chi$-profiles confirms that
the faults located in the Sierra Morena southern escarpment have experienced recent, although somehow diachronous, tectonic activity. Our analysis also evidences differences in relative uplift along this escarpment. Moreover, divide migration controls the deviation observed between modelled and observed knickpoints distribution as indicated by the computed values of the divide asymmetry index (DAI).

On the whole, our results deviate from those expected in a typical flexural forebulge. In addition to flexural related deformation,
the kinematic features of faults related to the current Sierra Morena relief point to Quaternary, strongly partitioned, transpressional deformation of the Betic foreland. This is compatible with the intraplate propagation of the Betic fold and thrust deformation.

**Code and data availability**

Codes rely on TopoToolbox (Schwanghart and Scherler, 2014), which is freely available in
https://topotoolbox.wordpress.com/download

TanDEM-X data were kindly provided by the German Aerospace Center (DLR) (project ID: DEM_HYDR3360); https://tandemx-science.dlr.de.

The software used for stereoplots is Faultkin (Richard W. Allmendinger © 2020-2022), which is freely available in https://www.rickallmendinger.net/

**Author contribution**

IE led the research and wrote the paper draft. AJB and MD made the quantitative geomorphological analyses. IE, AJB, JLY, JCB and FM participate in the fieldwork for both qualitative geomorphological analysis and structural analysis. All the authors participated in the results interpretation as well as in the text improvement.

**Competing interest**

The authors declare that no competing interests are present.



**Acknowledgements**

This research is part of projects PGC2018-100914-B-I00, funded by the Ministerio de Ciencia e Innovacíon (Spanish Government)/ AEI/ 10.13039/501100011033/ ERDF, and UPO-1259543, funded by the Consejería de Economía, Conocimiento, Compañías y Universidad (Andalusian Government)/ ERDF.

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
