# Peer review of "Geomorphic signature of relief rejuvenation in Sierra Morena (Betic forebulge, Spain): evidence of segmented uplift in a strongly strain-partitioned, tectonic scenario"

_Earth Surface Dynamics, 2022_

## Referee Comment (RC2)

**Review of 'Geomorphic signature of relief rejuvenation in Sierra Morena (Betic forebulge, Spain): evidence of segmented uplift in a strongly strain-partitioned, tectonic scenario' by Immaculada Expósito et al**

Reviewer: Colin P. Stark

July 7, 2022

**Summary comments**

The focus of this paper is part of the "forebulge" area of the Betics and the goal is to find geomorphic responses to ongoing tectonics using geomorphometric methods.

Presentation is to a high standard: the paper is well written, clearly structured, with good graphics, and is thoroughly referenced. The methods employed are a mix of old-school geomorphometry and more modern $\chi$-space analysis, combined with some field observations and GIS processing. It's a solid piece of work.

The hypothesis driving the study is that Neogene-present compression will manifest as a geomorphic response to relative uplift that is localized by strain partitioning and by reactivation of existing structures.

The methods employed, and the underlying philosophy, have a traditional flavour throughout, although the Perron/Royden $\chi$ tool is a relatively recent innovation. The key concept espoused by the authors is the old-school idea that landscapes change, and become "rejuvenated" in some sense, because of vertical motions driven by tectonics: signs of such rejuvenation include knickpoint formation and drainage divide motion. The evidence they present is purely morphometric; they argue it confirms their main hypothesis.

Although I am a little uncomfortable with this philosophy and with the classical-style quantitative methods, I strongly encourage publication. This paper represents a substantial body of work that contributes to our knowledge of the tectonic geomorphology of the Betic foreland. While it doesn't make any conceptual advances, it does present new data on this landscape and constraints on its recent evolution. It serves as a good example of the application of geomorphometric tools young and old, and (in my mind) illustrates their limitations.

**Specific comments**

**Abstract**

Well written, fairly concise, covering the context, goals, methods, results, and speculations.

**1. Introduction**

Describes the tectonic geomorphology of the Sierra Morena (as a flexural forebulge) largely in the context of Cenozoic deformation of the Betics. Detailed and apparently comprehensive.

It's here that the authors state their basic hypothesis, which is that "relief rejuvenation" in the SM is the result of Neogene-present faulting and folding. This is perfectly reasonable.

p.2, lines 35, 49, 51, 44: Typos in Cloetingh references, both here and elsewhere in the text.

Fig.1: typo "extracted from the [m] DEM". Component map (c) is in colour, but the hue (green only) carries no real information - it's all in the shade. I recommend simplifying to a grey-scale map instead.

**2. Geological setting and regional topographic features**

Review of the regional tectonics and topography. I am not familiar with this area or this literature, but this section comes across as thorough and scholarly.

Fig.2: Nice, clear map. All the graphics in this paper are of a very high standard. Advice: the sentence "The altitude base is the 12m..." need only be given in full in the first figure citation; after that, a cross-reference may suffice.

**3. Approach and method**

This section details the analytical methods used, including geomorphometric indexes such as mountain-front sinuosity, valley floor-to-height ratio, and drainage divide asymmetry. There is an explanation of RS/DEM-based knickpoint mapping and modeling, and $\chi$ transformation.

While I am not particularly keen on the old-style tools of geomorphometry (because they are rarely process-based, and are tough to relate to any underlying physics), they are considered to have value. The more modern Perron/Royden tool of $\chi$-space analysis is, however, founded on a process model, and it's the logical choice given the goals of the paper. I found some issues with the explanation of the method though.

p.9, lines 230-231: Lague calls the stream-power law the "Stream-Power Incision Model (SPIM)", which is a good name (for one thing, it's accurate) - so why not stick with that instead of writing "stream-power law (SPL) equation system" (which is a bit misleading).

p.9, line 235: typo "an"

p.9, line 236-241: There are some issues with this explanation of the $\chi$ idea. (1) Eqn.3 should use partial derivatives e.g. $\partial z/\partial t$. (2) Uplift rate $U$ is assumed constant with space and time in the integration in eqn.5, so it cannot be allowed to vary with both $x$ and $t$ as given in eqn.3. (3) The exponent $n$ is taken to be $n = 1$, which is standard practice although tough to justify, but the authors associate it with linearization ("linearises the SPL form ($n = 1$)), which is not why it's chosen or how linearization arises. (4) The assumptions involved are not clearly explained.

These shortcomings should be easy to fix. To help, let me be more clear how I see the steps and assumptions here - by writing out the equations again but with some subtle corrections that make explicit those assumptions. Taking the SPIM for the bedrock incision rate in channels in a region subject to a vertical rock uplift rate $U(x,t)$ in the frame whose elevation above some datum is given by $z(x,t)$, the rate of channel elevation change is given by:

$$\frac{\partial z(x,t)}{\partial t} = U(x,t) - KA^m(x) \left| \frac{\partial z(x,t)}{\partial x} \right|^n \tag{1}$$

where erodibility $K$ is assumed to be constant, and the upstream area at each point $x$ is constant with time $A(x)$. In the literature (e.g. Fox et al, 2014), the exponent $n$ is generally assumed to be $n = 1$ for simplicity (and for no other reason really!), while the exponent $m$ is observed to be around $1/3$ to $1/2$ but can be inferred from each data set. In reality, the ratio $m/n$ is inferred - which is what the authors do in section 6. Now, if (a big if) we assume steady-state balance between vertical tectonic uplift and erosion (notice we have assumed not horizontal tectonic deformation, which runs counter to the overall tectonic context of the Sierra Morena), we can write (notice the partials become full)

$$KA^m(x) \left| \frac{dz(x)}{dx} \right|^n = U \tag{2}$$

with $U$ and $z$ considered constant in $x$: $x$ dependence in $U$ would make integration problematic, while $t$ dependence would mean the equation would remain a partial differential equation and therefore not simply integrable:

$$\frac{dz(x)}{dx} = \left( \frac{U}{KA^m(x)} \right)^{1/n} = \left( \frac{U}{KA_0^m} \right)^{1/n} \left( \frac{A(x)}{A_0} \right)^{-m/n} \tag{3}$$

where the sign of the gradient has been ignored. This gives the integration

$$z(x) = \left( \frac{U}{KA_0^m} \right)^{1/n} \int_{x_b}^{x} \left( \frac{A(x')}{A_0} \right)^{-m/n} dx' \tag{4}$$

which is effectively what's given in eqns.4 and 5. My point here is that $U$ cannot be allowed to vary with $x$, otherwise it cannot be extracted from the integral. Nor can we legitimately allow $U$ to vary with time, because elevation has been assumed to be time-invariant $z(x)$. What's more, upstream area $A(x)$ does not, as written, vary with time either, which means that catchments areas can't grow or shrink and drainage divides can't move. Finally, erodibility $K$ must be the same everywhere.

These assumptions are not consistent with the very inferences typically made using the $\chi$ method, such as a history of changes in uplift rate, formation and propagation of knickpoints, and divide migration. In other words, the targets of $\chi$ analysis seem to be phenomena that the $\chi$ model precludes.

The logical escape route, at least as I understand it, is to say that the $\chi$ transformation provides a kind of null hypothesis (formation of channel profile subject to the strict assumptions above) against which a real profile can be compared. This is perfectly fine, and is the standard thinking (I believe), so I am not criticizing the authors for issues I associate with the method. Rather, I would prefer to see the logic of the $\chi$ explained better - either as I have tried to do, or in some other way if my understanding is incorrect. Specifically, would the authors comment on how they reconcile their knowledge of spatial variations on rock type (erodibility) and tectonic deformation with model constancy in $K$ and $U$?

p.10 line 241: should be $x_b$ not $xb$

p.10 line 256: capitalize "Gaussian"

p. 10 line 266: "with [the] Topo ordering scheme" - is this the name of the algorithm?

**4. Signs of relief rejuvenation in Sierra Morena**

Some qualitative geomorphological analysis using a mix of field observations, GIS mapping, and simple remote sensing. Geomorphic indexes and hypsometry are discussed. The work here is solid. I particularly like the observation of incised meanders, which (in my experience) can be a good indicator of local "uplift". It would have been nice to seem more description of this and other features that are mentioned only in passing and illustrated only with a blurred GE image.

A minor comment: here and elsewhere the indexes are sometimes written Vf, Smf, etc, and sometimes as $V_f$, $S_{mf}$. I would prefer to see subscripts used throughout.

**5. Structures related to relief segmentation and rejuvenation**

Structural fabrics. These are useful data to have, and their interpretation looks fine.

**6. Knickpoint pattern analysis**

An analysis of knickpoints using topographic profiles, both raw and $\chi$-transformed. Careful attention is given to the distorting effect of artificial dams. Observations are made regarding overall convexity or concavity of profiles in each catchment S1-S6: the strong up-convexity of S2 and S6 is much more clear in the $\chi$-transformed profiles.

p.18 table 2: The values of $m/n$ are incorrectly given as negative numbers. Both exponents are strictly positive, so this is probably just a misreading of eqn.5.

p.18 line 427: "S3 and S4 plots are seen to display a concave-up shape" - I don't see this. They look pretty straight to me, at least in the $\chi$-transformed profiles.

Fig. 8: Should be "Tributary inlets into reservoirs". And why are there two different symbols (empty circles and full diamonds) for these?

Fig. 8: I would prefer to see a specific explanation in this figure caption for how I'm supposed to interpret the figures. There is a lot of information here, but the caption provides no guidance as to what it's all supposed to mean.

Fig. 8: It would help if each graph pair were labeled with the name of the catchments to which they belong. I take it S1 = (a), (b); S2 = (c), (d), etc, but the caption is too cryptic and the subfigures themselves have no information. Readers shouldn't have to work to understand such fundamentals: they should "pop" out of the graphics themselves.

p.21 line 457: "show a strong consistency between the model and the location of a robust number of actual knickpoints" - what model? Having read this far, I'm still not clear on what model I should have in mind here.

**7. Discussion**

Speculation about the structural origins of the geomorphic features observed in the body of the paper, and on the tectonic origins of these structural influences.

p.23 fig 9: Nice plot of $\chi$ vs elevation, but it lacks any explanation as to how to interpret it. Explanation is given on p.22 lines 507-517 "the slope of plots can be interpreted as a proxy for [uplift rate]" - I suggest adding it to the caption here.

p.25 line 558: typo: "regards" should be "regard"

**References**

Typo on line 705

---

## Author Response (AR1)

We sincerely appreciate the constructive comments from both Stefan Hergarten and Colin P. Stark, which will surely contribute to improve our manuscript.

Please, find below our point-by-point response to both referees.

**in black reviewer comments**

in blue our response

in green changes in the manuscript

**REVIEWER 1 (Stefan Hergarten)**

This manuscript provides a geomorphic analysis of the Sierra Morena region in southern Spain. The main parts of the analysis are performed with the help of the TopoToolbox software. The observations from geomorphology are supported by structural geology.

Being not very familiar with the geology of Spain, the introduction (Sect. 1) and the description of setting and region (Sect. 2) were very useful for me. These sections are well written. This also holds for parts of the description of the methods (Sect. 3), except for the chi-transform (Sect. 3.2, see below).

However, my positive impression dropped when reading the results section (Sect. 4). First, I found it a bit hard to follow. As a serious problem, however, I feel that the results of the geomorphic analyses are rather non-unique or even somewhat weak. This does not mean that the conclusions are wrong, but I am not convinced that they can be drawn from the results in a solid way. Let me briefly explain my concerns about the different investigated topographic properties.

The referee is mainly concerned about the robustness of the presented geomorphic results to support our conclusions about relief rejuvenation. His doubts are mostly related to both geomorphic indexes results and knickpoints analysis, and he considers that this part needs to be strengthened (at the end of his major comments).

In general term, we agree that the mentioned analyses must be addressed very carefully since, as RC1 points, they do not offer unique interpretations when results are taken separately. However, we consider that all these results, taken together, share a common interpretation, being consistent with relief rejuvenation associated to recent tectonics. This common interpretation is reinforced by the presence in our study area of key geomorphic features, commonly used in this type of studies, to identify rejuvenation of previous flat, regional scale, paleoreliefs (in our case, an Upper Miocene peneplain shaped at the Guadalquivir foreland basin level).

These features are:

-preserved remnants of the mentioned Miocene peneplain on top of the Sierra Morena summits (Fig.1.b and c), now located several hundred meters above its relative base level (i.e., the Guadalquivir foreland basin). This paleo surface is at places stepped and/or tilted by faulting.

-patches of shallow-water Miocene deposits of the Guadalquivir foreland basin on top of the Sierra Morena escarpment, uplifted up to 400 m asl. As the peneplain, these patches are frequently stepped and tilted.

-numerous river knickpoints, not controlled by lithological contrasts, which are mainly located in the lower course of the river basin.

-good and abundant examples of incised meanders.

-river piracy and windgap formation upstream to an active fault trace.

We will emphasize the existence of these features in the Introduction as well as in the Summary and conclusions.

Once this overall consideration is made, we start out point by point response. We assume that our analyses require further justification (in the terms exposed below), mainly within sections 4, 6 and 7, in a revised version.

**MAJOR COMMENTS**

**(1) Smf values (Sect 4.2)**
These values just describe the shape of the mountain front and are not very meaningful in the context of relief juvenation.

We agree that it needs to be explained more accurately. This index alone does not test relief reactivation but potential recent activity of those faults coincident with mountain fronts. The Sierra Morena relief mostly consists of roughly flat, subhorizontal or gently SSE-ward dipping summits, separated by incised valleys. As mentioned before, these summits often exhibit either remnants of an Upper Miocene, relict peneplain or Upper Miocene rocks deposited in the Guadalquivir foreland basin. Therefore, these relief elements have been relatively uplifted from their initial position at the Guadalquivir basin after the Late Miocene. Given that we have discarded vertical faults and avoided lithological contrasts, straight mountain fronts (close to 1) indicate young, non-eroded escarpments. This indicates that the southern Sierra Morena relief has been rejuvenated by Quaternary activity of the escarpment-related faults. This index has been combined with Vs in several previous work as a proxy to test recent tectonic activity (e.g. Silva et al., 2003).

We will modify Section 4.2 to discuss the Smf values in the terms listed above.

**(2) Vf values (Sect. 4.2, Fig. 5a)**
The values shown in Fig. 5a just show that the valleys are deeply incised in the mountains and rather flat in the alluvial plain. This is not very surprising and immediately recognized in the topography.

In theory, the VF value in a graded long profile (mature rivers) tends to decrease gradually towards the river mouth along the river profile. This does not happen in the selected rivers, where values drop abruptly upstream from the fault-related mountain fronts. Additionally, values are frequently lower in the escarpment sector (quite close to the river mouth) than in higher reaches of the same river, thus suggesting that river incision in this sector responds to the slope disequilibrium located in the fault-related escarpment.

We will add in both Fig.5 and in Table 1 the VF values of these higher reaches to support our conclusions. Furthermore, these new values will be discussed in Section 4.2.

**(3) Hypsometric curves and HI values (Sect. 4.2, Fig. 5b+c)**
Some of the hypsometric curves are indeed convex upward. However, we have to be careful with the interpretation. The "default" S-shaped curve relies on the stream-power law being applicable to the entire area, so also for very small catchment sizes. As soon as hillslope processes come into play, the relief at small catchment sizes decreases, which also causes a convex upward shape. So interpreting these curves and the respective HI values is difficult. Comparing individual sub-catchments as shown in Fig. 5c may be helpful. However, the variation

among the sub-catchments seems to be somewhat unsystematic, although the location of the sub-catchments is not visible. So placing the rather tentative conclusions drawn here on solid ground would require a more thorough analysis.

We agree that our interpretation in this regard needs to be justified more thoroughly in section 7 of the manuscript. As the referee argues, previous works have pointed out that the basin hypsometry can be conditioned, among other factors, by the catchment size (e.g., Hurtrez et al., 1999, and references therein). Indeed, the small catchments hypsometry can be dominated by hillslope processes, leading to convex upward hypsometric curves. Nevertheless, our selected catchments are large scale catchments, being S1, S2, S5 and S6 larger than 1000 km2 (up to 1875 km2 in S1). Even S3 and S4, which are smaller, are of several hundred km2, i.e., of the same order that the "large" catchments analysed in the above-mentioned paper. Therefore, we can assume that our catchments are dominated by river processes and, therefore, convex upward curves and high HI values can be taken as a proxy for recent tectonics.
Regarding the sub-catchments of S2 (Fig.5c), they are small-scale basins (between 18 and 71 km2). S.2.4, S.2.5 and S.2.6 are located on the hanging-wall of the VFS, being therefore in a tectonic situation that is comparable to the area of S2 analysed in Fig. 5b (upstream of VFS). Although they are two orders of magnitude smaller, neither their curves are not more convex upward than the S2, nor their HI values are higher. This seems to indicate that the hypsometry of the catchments located on the VFS hanging wall is significantly conditioned by the Quaternary reactivation of such fault system.

To better discuss our results, we will include the size of the selected catchments in Table. 1, and will draw the S2 sub-catchments on Fig. 1b and Fig. 2a. The text will be modified to include the arguments listed above in sections 3.1 and 4.2.

**(4) Knickpoints (Sect. 6)**
The analysis of knickpoints seems to be a central component. However, the results are very non-unique here. I only recognize a clear clustering of knickpoints in the chi-plot only for catchment S2, so at the Viar fault system. For the other catchments, which are related to the Sierra Morena rupture, I do not recognize any clear evidence for a rejuvenation in the knickpoint pattern.

As stated above, the relief rejuvenation of Sierra Morena in our study area has been primarily inferred from key qualitative features that, although observed all along the Southern Sierra Morena, are particularly clear in S2 and S6. Regarding S6, there is a strong chain of evidence supporting that this rejuvenation is linked to recent tectonic activity. The responsible faults are distributed along the escarpment (the highest one in the entire study area) that limited Sierra Morena N and NE of the Medina Azahara archaeological site (Fig.4).

These features are:

- The Guadiato long-profile (S6) is remarkably convex upward and the great majority of the abundant knickpoints are in the lower course of the stream network, suggesting that relatively recent perturbations must have occurred nearby the escarpment.

-The Miocene peneplain (once at the base level of the foreland basin) as well as outcrops of the Guadalquivir basin sediments are located on the very top of the escarpment, several hundred meters above the Guadalquivir basin.

-The peneplain, regionally dipping toward the SSE, is tilted NW-ward in the hanging wall of the faults related to the escarpment, so that the SE divide of the Guadiato catchment coincides with the upper edge of the escarpment. Thus, this tilting is plausibly caused by such faulting.

-The uplift and tilting of the peneplain have dismantled the lower course of the Guadiato River, forcing the river to migrate, leaving a wind gap that currently plunges gently to the north.

-North of the wind gap, the Guadiato River (S6) describes an elbow due to stream piracy, being captured by a lower neiforghbour stream.

-The Guadiato River (S6) stream network exhibits multiple incised meanders.

-Medina Zahara ancient city, located at the very foot of the main escarpment, shows damages by earthquakes and has been buried by debris deposits in the last 1000 years, accounting for escarpment instability. Interestingly, an earthquake (magnitude 3,1) was mainly felt in the ancient city in 2004.
All these facts point to recent tectonic activity leading to the rejuvenation of the S6 basin relief.

Therefore, the reason for the imperfect knickpoints clustering that S6 shows in comparison to S2 must lie elsewhere.

As we suggest in the manuscript, the knickpoints dispersion on chi-profiles could partly occur by basin divides migration, as proposed by Schwanghart and Scherler (2021). These authors proved that processes relate to divide network reorganization alter drainage areas and discharge, thus impacting on knickpoint celerities. This fact can result in knickpoints scattering on chi-profiles. Additionally, the reason may be also related to the strain partitioning expected in a region widely affected by previous mechanical discontinuities. Indeed, knickpoints originated by the same perturbation (e.g., a single event in a single fault) must cluster on chi-space. This situation happens in S2 where deformation is very localized by a discrete fault system (VFS), here interpreted as a reactivated main Variscan boundary. However, in the rest of our study area, the recent deformation does not seem to be accommodated by major faults but highly partitioned into distributed, variably oriented, faults. This means that several perturbations originated in different faults could be superimposed along the stream network and, therefore, knickpoints would not cluster. In this regard, this geomorphic tool could be useful to test localization versus distribution of the strain.

This discussion will be integrated in Section 7.
Moreover, in order to better discuss the absence of a clear knickpoint cluster in S6 basin and to provide further evidence about relief rejuvenation, we will perform a swath profile along the divide among the Guadiato river basin and the minor basins close to the Medina Azahara archeological site. It will allow us to highlight the presence of a series of wind gaps. Such landforms are linked to the river capture described by the river elbow and the incised meanders. All this evidence strengthens the hypothesis of the relief rejuvenation also without a clear knickpoint cluster. Indeed, drainage basin contributing area have changed rapidly during the river capture affecting the knickpoint celerity and then the knickpoint expected location.

**(5) Drainage divide network asymmetry (DAI, Sect. 6)**
The results on the drainage divides are interesting. To my understanding, however, relief juvenation should rather affect the lower parts of the catchments than the drainage divides. So I would think that the DAI reflects the long-term activity rather than a juvenation.

The drainage divide network asymmetry was calculated not to demonstrate the relief juvenation rather to investigate possible inconsistencies between expected and observed knickpoint patterns. Indeed, as it is an index that describes the asymmetry of a divide based on the hillslope relief difference between the two sides, it can be useful to individuate areas where divide disequilibrium is occurring and to possibly correlate with anomalous knickpoint location Moreover, we performed the analysis for divides with an order greater than 10 (referring to Topo ordering scheme, where divide orders increase by 1 at each junction), so including minor internal divides too.
In fact, many knickpoints, especially those in the higher portions of some basins (see S1 and S5), have been affected by divide mobility. However, often minor internal divides in the main basins show asymmetry which could also explain the anomalous position of some knickpoints in the lower portion of the basin (see S1).

In order to support the use of DAI to detect actual moving divides, we will show in a new figure, together with the swath profile along the divide among the Guadiato river basin and the minor

basins close to the Medina Azahara archeological site, also a DAI profile. This will demonstrate that high DAI values are related to asymmetric divide that are still migrating.

In sum, I am not convinced that the results of the geomorphic analyses are strong enough to support the conclusions about relief rejuvenation. It might, however, be possible to strengthen this part. Finding clearer evidence in some properties might be sufficient. Provided that this is possible, a revised version would make sense, and I would be happy to review it.

We expect that all the changes listed above will contribute to strengthening our results and conclusions.

**MINOR COMMENTS**

In this case, some other points should also be addressed:

**Sect. 3.1**
I am not familiar with the valley floor-to-height ratio Vf. How is the width of the valley floor determined?

The valley floor width is determined by topographic cross-sections (extracted from the DEM), drawn perpendicular to the stream (e.g., Giaconia et al., 2012). Aerial photos are additionally used in alluvial stream segments where sediments help to determine the valley floor width.

**Sect. 3.2**
The description of the chi-transform is quite wrong. It is not a linearization and has nothing to do with n = 1. And Eq. (4) relies on specific conditions.

We agree with the comment.

We will correct accordingly, commenting the main assumptions of the approach, as also suggested by reviewer 2.
Here is the new text:

"For the river profile analysis, we applied the integral approach introduced by Perron and Royden (2013) of the Stream Power Incision Model (SPIM, Lague, 2014). According to the SPIM, the evolution of the river profile is described as the change through time $t$ in elevation $z$ of a channel point located to an $x$ upstream horizontal distance, which can be calculated using the expression:

$$\frac{\delta z(x,t)}{\delta t} = U(x,t) - K(x,t)\, A\,(x,t)^{\,m} \left|\frac{\delta z}{\delta x}\right|^{n}, \tag{3}$$

where $U$ is the uplift, A is drainage area, $m$ and n are positive an empirical constants and $K$ is the erodibility.

If both processes are perfectly balanced, a state of a dynamic equilibrium or steady state ($\delta z/\delta t = 0$) is assumed. In this case, the Equation 3 can be rearranged as:

$$\left|\frac{dz}{dx}\right| = \left(\frac{U}{K}\right)^{\frac{1}{n}} A(x)^{-\frac{m}{n}} \tag{4}$$

where U and K do not vary in time and in space and drainage configurations remain unchanged. Equation 4 predicts a power-law relationship between slope and drainage area A (Hack, 1957).

Perron and Royden (2013) proposed a transformation of the horizontal spatial coordinates which linearizes the power-law relation, by the integration of $dz/dx$ with respect to $x$ from a base level $x_b$ to an arbitrary upstream point $x$ of the channel.

$$\int \frac{dz}{dx}\, dx = z(x_b) + \left(\frac{U}{KA(x)^m}\right)^{\frac{1}{n}} \int \frac{A_0}{A(x)^{m/n}} \tag{5}$$

where $A_0$ is an arbitrary scaling area. Performing the integration, Equation 5 becomes:

$$z(x) = z(x_b) + \left(\frac{U}{KA_0{}^m}\right)^{\frac{1}{n}} \chi, \qquad\qquad (6a)$$

with

$$\chi = \int_{x_b}^{x} \left(\frac{A_0}{A(x)}\right)^{\frac{m}{n}} dx \qquad\qquad (6b)$$

Even if a river is not in a topographic steady state, the advantage of using the χ plot of its longitudinal profile is that transient signals with a common origin (e.g., fault-related knickpoints) propagating upstream through different channels, along either the main stem or tributaries, plot in the same location in transformed coordinates χ and z (Perron and Royden, 2013; Schwanghart and Scherler, 2020). Additionally, in steady state conditions, the main stem and tributaries become aligned once the best *m/n* value has been determined. Thus, transient signals are easier to identify when the transformed profiles deviate from such trend. Thus, χ serves as a metric for distances travelled by perturbations upstream in the river network (Fox et al., 2014). For this reason, we consider as base level $x_b$ the intersection between streams and fault traces. Furthermore, to find the best *m/n* value for each basin, i.e., with the best-fit linear relationship between *χ* and elevation (Perron and Royden, 2013), we used the TopoToolbox "mnoptimvar" and "robustcov" functions (Schwanghart and Scherler, 2014).

Finally, according to Eq. (6a), the slope of χ-profiles is directly proportional to the uplift rate (U) and inversely proportional to the erodibility (K). In this regard, we assume that there are no strong lithological contrasts in the study area, justifying that the lithological control on fluvial erosion is quite homogeneous (K constant). Then, the slope of χ plots can be interpreted as a proxy of the tectonic activity. In this sense, the uplift U can be considered constant for each analysed basin, since each one depends on a different fault and the upstream area can be assumed as an uplifted block."

**Sect. 6**
Why are the m/n ratios in Table 2 negative? This does not make sense. And I guess that all considered catchments are similar concerning their fluvial erosion characteristics. So fitting different m/n ratios for the individual catchments is also not very useful. In particular, the small ratios of S3 and S4 seem to be an artifact. So we should either fit one common value to all catchments or -- if we want a simple solution -- say that the big catchments are consistent with the widely used reference value m/n = 0.45. And maybe it would be clearer to use the concavity index (theta) instead of m/n.

As also pointed out by reviewer 2, negative m/n ratios are misreading of eqn. 5. Regarding the m/n ratios, we decided to keep the individual catchment values because they themselves give us information on relief juvenation.

We will correct the signs. We will integrate a new figure in which we report the chi profiles that have the lowest chi disorder, intended as the variability of elevation values for similar values of chi (Hergarten et al., 2016).

References:

Giaconia, F.; Booth-Rea, G.; Martínez-Martínez, J.M; Azañón, J.M; Pérez-Peña, J.V.; Pérez-Romero, J., and Villegas, I.: Geomorphic evidence of active tectonics in the Sierra Alhamilla (eastern Betics, SE Spain), Geomorphology, 145-146, 90-106, https://doi.org/10.1016/j.geomorph.2011.12.043, 2012.

Hergarten, S., Robl, J., and Stüwe, K.: Tectonic geomorphology at small catchment sizes – extensions of the stream-power approach and the χ method, Earth Surf. Dynam., 4, 1–9, https://doi.org/10.5194/esurf-4-1-2016, 2016.

Hurtrez, J.E., Sol, C. and Lucazeau, F.: Effect of drainage area on hypsometry from an analysis of small-scale drainage basins in the Siwalik Hills (Central Nepal), Earth Surf. Proc. Land., 24, 799-808, https://doi.org/10.1002/(SICI)1096-9837(199908)24:9<799::AID-ESP12>3.0.CO; 2-4, 1999.

Schwanghart, W. and Scherler, D.: Divide mobility controls knickpoint migration on the Roan Plateau (Colorado, USA), Geology, 48(7), 698-702, https://doi.org/10.1130/G47054.1, 2020.

Silva, P.G., Goy, J.L., Zazo, C., and Bardaí, T.: Fault-generated mountain fronts in southeast Spain: geomorphologic assessment of tectonic and seismic activity, Geomorphology, 50, 203–225, https://doi.org/10.1016/S0169-555X(02)00215-5, 2003.

**REVIEWER 2 (Colin P. Stark)**

**SUMMARY COMMENTS**

The focus of this paper is part of the "forebulge" area of the Betics and the goal is to find geomorphic responses to ongoing tectonics using geomorphometric methods.

Presentation is to a high standard: the paper is well written, clearly structured, with good graphics, and is thoroughly referenced. The methods employed are a mix of old-school geomorphometry and more modern c-space analysis, combined with some field observations and GIS processing. It's a solid piece of work.

The hypothesis driving the study is that Neogene-present compression will manifest as a geomorphic response to relative uplift that is localized by strain partitioning and by reactivation of existing structures.

The methods employed, and the underlying philosophy, have a traditional flavour throughout, although the Perron/Royden χ tool is a relatively recent innovation. The key concept espoused by the authors is the old-school idea that landscapes change, and become "rejuvenated" in some sense, because of vertical motions driven by tectonics: signs of such rejuvenation include knickpoint formation and drainage divide motion. The evidence they present is purely morphometric; they argue it confirms their main hypothesis.

Although I am a little uncomfortable with this philosophy and with the classical-style quantitative methods, I strongly encourage publication. This paper represents a substantial body of work that contributes to our knowledge of the tectonic geomorphology of the Betic foreland. While it doesn't make any conceptual advances, it does present new data on this landscape and constraints on its recent evolution. It serves as a good example of the application of geomorphometric tools young and old, and (in my mind) illustrates their limitations.

**SPECIFIC COMMENTS**

**Abstract**
Well written, fairly concise, covering the context, goals, methods, results, and speculations.

**1. Introduction**

Describes the tectonic geomorphology of the Sierra Morena (as a flexural forebulge) largely in the context of Cenozoic deformation of the Betics. Detailed and apparently comprehensive.

It's here that the authors state their basic hypothesis, which is that "relief rejuvenation" in the SM is the result of Neogene-present faulting and folding. This is perfectly reasonable.

p.2, lines 35, 49, 51, 44: Typos in Cloetingh references, both here and elsewhere in the text. Fig.1: typo "extracted from the [m] DEM". Component map (c) is in colour, but the hue (green only) carries no real information - it's all in the shade. I recommend simplifying to a grey-scale map instead.

**2. Geological setting and regional topographic features**
Review of the regional tectonics and topography. I am not familiar with this area or this literature, but this section comes across as thorough and scholarly.

Fig.2: Nice, clear map. All the graphics in this paper are of a very high standard. Advice: the sentence "The altitude base is the 12m. . . " need only be given in full in the first figure citation; after that, a cross-reference may suffice.

We agree with the comment.

We will correct it accordingly.

**3. Approach and method**
This section details the analytical methods used, including geomorphometric indexes such as mountain-front sinuosity, valley floor-to-height ratio, and drainage divide asymmetry. There is an explanation of RS/DEM-based knickpoint mapping and modeling, and χ transformation.

While I am not particularly keen on the old-style tools of geomorphometry (because they are rarely process-based, and are tough to relate to any underlying physics), they are considered to have value. The more modern Perron/Royden tool of χ-space analysis is, however, founded on a process model, and it's the logical choice given the goals of the paper. I found some issues with the explanation of the method though.

p.9, lines 230-231: Lague calls the stream-power law the "Stream-Power Incision Model (SPIM)", which is a good name (for one thing, it's accurate) - so why not stick with that instead of writing "stream-power law (SPL) equation system" (which is a bit misleading).

We agree with the comment.

We will correct it accordingly.

p.9, line 235: typo "an"

We agree with the comment.

We will correct it accordingly.

p.9, line 236-241: There are some issues with this explanation of the χ idea. (1) Eqn.3 should use partial derivatives e.g. $\delta z/\delta t$. (2) Uplift rate U is assumed constant with space and time in the integration in eqn.5, so it cannot be allowed to vary with both x and t as given in eqn.3. (3) The exponent n is taken to be n = 1, which is standard practice although tough to justify, but the authors associate it with linearization ("linearises the SPL form (n = 1)), which is not why it's chosen or how linearization arises. (4) The assumptions involved are not clearly explained.

These shortcomings should be easy to fix. To help, let me be more clear how I see the steps and assumptions here - by writing out the equations again but with some subtle corrections that make explicit those assumptions. Taking the SPIM for the bedrock incision rate in channels in a

region subject to a vertical rock uplift rate U(x, t) in the frame whose elevation above some datum is given by z(x, t), the rate of channel elevation change is given by:

$$\frac{dz(x,t)}{dt} = U(x,t) - K A (x)^m \left| \frac{dz(x,t)}{dx} \right|^n$$

where erodibility K is assumed to be constant, and the upstream area at each point x is constant with time A(x). In the literature (e.g. Fox et al, 2014), the exponent n is generally assumed to be n = 1 for simplicity (and for no other reason really!), while the exponent m is observed to be around 1/3 to 1/2 but can be inferred from each data set. In reality, the ratio m/n is inferred - which is what the authors do in section 6. Now, if (a big if) we assume steady-state balance between vertical tectonic uplift and erosion (notice we have assumed not horizontal tectonic deformation, which runs counter to the overall tectonic context of the Sierra Morena), we can write (notice the partials become full)

$$K A (x)^m \left| \frac{dz(x)}{dx} \right|^n = U$$

with U and z considered constant in x: x dependence in U would make integration problematic, while t dependence would mean the equation would remain a partial differential equation and therefore not simply integrable:

$$\frac{dz(x)}{dx} = \left( \frac{U}{KA^m(x)} \right)^{\frac{1}{n}} = \left( \frac{U}{KA_0{}^m} \right)^{\frac{1}{n}} \left( \frac{A(x)}{A_0} \right)^{-\frac{m}{n}}$$

where the sign of the gradient has been ignored. This gives the integration

$$z(x) = \left( \frac{U}{KA_0{}^m} \right)^{\frac{1}{n}} \int_{xb}^{x} \left( \frac{A_0}{A(x')} \right)^{-\frac{m}{n}} dx'$$

which is effectively what's given in eqns.4 and 5. My point here is that U cannot be allowed to vary with x, otherwise it cannot be extracted from the integral. Nor can we legitimately allow U to vary with time, because elevation has been assumed to be time-invariant z(x). What's more, upstream area A(x) does not, as written, vary with time either, which means that catchments areas can't grow or shrink and drainage divides can't move. Finally, erodibility K must be the same everywhere.

These assumptions are not consistent with the very inferences typically made using the χ method, such as a history of changes in uplift rate, formation and propagation of knickpoints, and divide migration. In other words, the targets of χ analysis seem to be phenomena that the χ model precludes.

The logical escape route, at least as I understand it, is to say that the χ transformation provides a kind of null hypothesis (formation of channel profile subject to the strict assumptions above) against which a real profile can be compared. This is perfectly fine, and is the standard thinking (I believe), so I am not criticizing the authors for issues I associate with the method. Rather, I would prefer to see the logic of the χ explained better - either as I have tried to do, or in some other way if my understanding is incorrect. Specifically, would the authors comment on how they reconcile their knowledge of spatial variations on rock type (erodibility) and tectonic deformation with model constancy in K and U?

We agree with the comment.

We will correct it as reported in the answer to the comment 3.2 of reviewer 1.

p.10 line 241: should be xb not xb

We agree with the comment.

We will correct it accordingly.

p.10 line 256: capitalize "Gaussian"

We agree with the comment.

We will correct it accordingly.

p. 10 line 266: "with [the] Topo ordering scheme" - is this the name of the algorithm?

Yes, it is.

We will explicit it accordingly.

**4. Signs of relief rejuvenation in Sierra Morena**
Some qualitative geomorphological analysis using a mix of field observations, GIS mapping, and simple remote sensing. Geomorphic indexes and hypsometry are discussed. The work here is solid. I particularly like the observation of incised meanders, which (in my experience) can be a good indicator of local "uplift". It would have been nice to seem more description of this and other features that are mentioned only in passing and illustrated only with a blurred GE image.

We agree with the comment.

We will add a slope/topographic map to Fig.3 where the geometry of some clearly incised meanders can be observed. Accordingly, we will describe them better in Section 4.1.

A minor comment: here and elsewhere the indexes are sometimes written Vf, Smf, etc, and sometimes as Vf , Smf . I would prefer to see subscripts used throughout.

We agree with the comment.

We will correct it accordingly.

**5. Structures related to relief segmentation and rejuvenation**
Structural fabrics. These are useful data to have, and their interpretation looks fine.

**6. Knickpoint pattern analysis**
An analysis of knickpoints using topographic profiles, both raw and χ-transformed. Careful attention is given to the distorting effect of artificial dams. Observations are made regarding overall convexity or concavity of profiles in each catchment S1-S6: the strong up-convexity of S2 and S6 is much clearer in the χ-transformed profiles.

p.18 table 2: The values of m/n are incorrectly given as negative numbers. Both exponents are strictly positive, so this is probably just a misreading of eqn.5.

We agree with the comment.

We will correct it accordingly.

p.18 line 427: "S3 and S4 plots are seen to display a concave-up shape" - I don't see this. They look pretty straight to me, at least in the χ-transformed profiles.

We were referring to the longitudinal profiles and not to χ-transformed profiles. Indeed, also the computed mn ratios confirm the description.

We will clarify the description reference to the longitudinal profile, as follows:
 "Despite these dam-related interruptions, the S3 and S4 longitudinal profiles seems to display a slightly concave-up shape, whereas those corresponding to S2 and S6 show a conspicuous convex-up geometry."

Fig. 8: Should be "Tributary inlets into reservoirs". And why are there two different symbols (empty circles and full diamonds) for these?

We agree with the comment.

We will correct it accordingly.

Fig. 8: I would prefer to see a specific explanation in this figure caption for how I'm supposed to interpret the figures. There is a lot of information here, but the caption provides no guidance as to what it's all supposed to mean.

We agree with the comment.

We will correct it accordingly.
The new caption will be as follows (figure numbers are changed in the revised version):
"Fig. 9. Knickpoint distribution along S1-S6 stream networks. (a), (c), (e), (g), (i) and (k) river longitudinal profiles; (b), (d), (f), (h), (j), and (l) χ transformed longitudinal profiles. The covariance estimates for knickpoints, calculated on the χ transformed distance from the outlet (spatial covariate) with and without dams, indicate the probability of finding knickpoints along the transformed profile. Knickpoints, colored by elevation and sized by height, artificial dams, and tributary inlets into the reservoirs are projected on profiles."

Fig. 8: It would help if each graph pair were labeled with the name of the catchments to which they belong. I take it S1 = (a), (b); S2 = (c), (d), etc, but the caption is too cryptic and the subfigures themselves have no information. Readers shouldn't have to work to understand such fundamentals: they should "pop" out of the graphics themselves.

We agree with the comment.

We will correct it accordingly.

p.21 line 457: "show a strong consistency between the model and the location of a robust number of actual knickpoints" - what model? Having read this far, I'm still not clear on what model I should have in mind here.

We agree with the comment. The model is intended to be the one of expected knickpoint patterns.

We will correct it accordingly.
Here is the new text:
"show a strong consistency between the probability of having knickpoints and the actual location of a robust number of knickpoints."

**7. Discussion**
Speculation about the structural origins of the geomorphic features observed in the body of the paper, and on the tectonic origins of these structural influences.

p.23 fig 9: Nice plot of χ vs elevation, but it lacks any explanation as to how to interpret it. Explanation is given on p.22 lines 507-517 "the slope of plots can be interpreted as a proxy for [uplift rate]" - I suggest adding it to the caption here.

We agree with the comment. The model is intended the one of expected knickpoint patterns.

We will correct it accordingly.
Here is the new caption:
Fig.10. χ versus elevation plot of knickpoint and artificial dams related to each drainage basin. The linear fits and the associated regression coefficients with and without artificial dams are reported. According to Eq. (6), the slope of χ-profiles is directly proportional to the uplift rate (U) and inversely proportional to the erodibility (K). Since there are no strong lithological contrasts in the study area, the slope of plots can be interpreted as a proxy of the tectonic activity.

p.25 line 558: typo: "regards" should be "regard"

We agree with the comment.

We will correct it accordingly.

**References**
Typo on line 705

We agree with the comment.

We will correct it accordingly.

---

## Author Response (AR2)

**in black reviewer comments**

**in blue our response**

**in green changes in the manuscript**

**# REVIEWER 1 (Stefan Hergarten)**

Dear Authors,

overall, I am quite satisfied with your revisions. Nevertheless, there are a few lingering issues that should be seriously taken into account.

(1) The description of the chi-method is much better now. However, three things still need to be repaired:
(a) Equation (5) is still not correct: The denominator in the parentheses must be K A_0^m instead of K A(x)^m (so the same as in Eq. 6a) and the integral must be exactly the same as in Eq. 6b).

We corrected it.

$$\int \frac{dz}{dx} dx = z(x_b) + \left( \frac{U}{KA_0{}^m} \right)^{\frac{1}{n}} \int_{x_b}^{x} \left( \frac{A_0}{A(x)} \right)^{\frac{m}{n}} dx$$

(b) The statement that all channels "plot in the same location in transformed coordinates chi and z" even if a river is not in a steady state is a bit misleading. In a transient state or under spatially heterogenous conditions, knickpoints are only located at the same chi values, while the z values may differ.

We corrected it.

Even if a river is not in a topographic steady state, the advantage of using the χ plot of its longitudinal profile is that transient signals with a common origin (e.g., fault-related knickpoints), propagating upstream through different channels, along either the main stem or tributaries, plot in the same location in transformed coordinate χ (Perron and Royden, 2013; Schwanghart and Scherler, 2020).

(c) The term "proportional" typically refers to linear relations. So the proportionality of the slope to the uplift rate and the inverse proportionality to the erodibility only holds for n = 1. In order to justify the usage of the chi-slope as a proxy for the uplift rate, however, you do not need the proportionality in the strict sense. So you can fix it by rewording and keep it valid for n not equal to 1. Another little point is that the dependence of the chi-slope on the uplift rate relies on (local) equilibrium conditions. In general, the chi-slope is a proxy for the actual erosion rate, which can be transferred to uplift rates only for equilibrium conditions.

We corrected it.

Finally, according to Eq. (6a), the slope of χ-profiles is dependent to the uplift rate and the erodibility: it increases as the uplift rate (U) increases and decreases as the erodibility (K) increases. In general, the chi-slope is a proxy for the actual erosion rate, which can be transferred to uplift rates only for equilibrium conditions.

(2) Sect. 6: In my opinion, it would still be better to use the same m/n ratio (= concavity index theta) for all considered catchments (perhaps 0.45 after discussing the variations in theta obtained for the catchments). There is no reason why the erosional environment should differ

much among the catchments, which would justify different concavities. Using different m/n ratios, we cannot compare chi-values and thus the locations of knickpoints across catchments. (e.g., lines 485-491). In each case, it should be taken into account that the apparent concavity of S3 and S4 (lines 477-488) is probably owing to the low concavity index.

We corrected it. Accordingly, figs. 9, 10 and 11 have been modified.

To perform χ transformations of longitudinal river profiles, we first have calculated for each basin the m/n ratio in Eq. (6) that minimises the variability of elevation values for similar values of χ (Fig. 8). The obtained values indicate that the larger catchments (S1, S2, S5, S6) are consistent with the widely used reference value m/n = 0.45, while the smaller ones (S3, S4) show lower values. Since all considered catchments are similar concerning their fluvial erosion characteristics and in order to compare χ -values and thus the locations of knickpoints across catchments, a reference value m/n = 0.45 was used to all the catchments.

(3) Fig. 8: Use m/n or theta instead of "mn" and remove "with" in caption.

We corrected it.

Best regards,

Stefan Hergarten

**REVIEWER 2 (Colin P. Stark)**

The authors have made a solid effort to respond to the reviewers' comments and have revised accordingly. I am happy to recommend publication.

Some technical corrections:

p13, eqn 3 and text on line 46: please don't use δ (one kind of lower-case delta), which means variational or substantial derivative in math/physics; instead use the partial differential symbol ∂ (another kind of lower-case delta)

We corrected it.

$$\frac{\partial z(x,t)}{\partial t} = U(x,t) - K(x,t) \, A\,(x,t)^{\,m} \left|\frac{\partial z}{\partial x}\right|^{\,n},$$  (3)

where U is the uplift, A is the drainage area, m and n are positive an empirical constants and K is the erodibility. If both processes are perfectly balanced, a state of a dynamic equilibrium or steady state ($\partial z/\partial t = 0$) is assumed.

p16, fig 2(b): typo "insiced"

We are afraid that the term "insiced" does not appear on Fig.2(b), but only on Fig. 3(d)

p17, fig 3(d): typo "insiced"

We corrected it.

p29, fig 8: "mn" should be "m/n" on all subfigs

We corrected it.

p41, l.699: should be "localised"

We corrected it.

---

## Author Response (AR3)

Dear Editor:

Thank very much for your final decision. We are very glad to publish our work in Earth Surface Dynamics and hope that your readers find it interesting.

As you recommended to us, we have changed the tittle of the article to make it shorter and more direct. Your proposal is perfect for us since it keeps all the key words. So that, thanks again for the tittle.

Best regards